# LncRNA *VEAL2* regulates PRKCB2 to modulate endothelial permeability in diabetic retinopathy

Paras Sehgal[1,2] (ID), Samatha Mathew[1,2], Ambily Sivadas[1,2], Arjun Ray[1,2,†], Jyoti Tanwar[1,2,3] (ID), Sushma Vishwakarma[4], Gyan Ranjan[1,2] (ID), K V Shamsudheen[1,2], Rahul C Bhoyar[1], Abhishek Pateria[1], Elvin Leonard[1], Mukesh Lalwani[1], Archana Vats[1], Rajeev R Pappuru[5], Mudit Tyagi[5], Saumya Jakati[4], Shantanu Sengupta[1,2], Binukumar B K[1,2] (ID), Subhabrata Chakrabarti[4] (ID), Inderjeet Kaur[4], Rajender K Motiani[3], Vinod Scaria[1,2,*] (ID) & Sridhar Sivasubbu[1,2,**] (ID)

## Abstract

Long non-coding RNAs (lncRNAs) are emerging as key regulators of endothelial cell function. Here, we investigated the role of a novel vascular endothelial-associated lncRNA (*VEAL2*) in regulating endothelial permeability. Precise editing of *veal2* loci in zebrafish (*veal2*[gib005Δ8/+]) induced cranial hemorrhage. *In vitro* and *in vivo* studies revealed that *veal2* competes with diacylglycerol for interaction with protein kinase C beta-b (Prkcbb) and regulates its kinase activity. Using PRKCB2 as bait, we identified functional ortholog of *veal2* in humans from HUVECs and named it as *VEAL2*. Overexpression and knockdown of *VEAL2* affected tubulogenesis and permeability in HUVECs. *VEAL2* was differentially expressed in choroid tissue in eye and blood from patients with diabetic retinopathy, a disease where PRKCB2 is known to be hyperactivated. Further, *VEAL2* could rescue the effects of PRKCB2-mediated turnover of endothelial junctional proteins thus reducing hyperpermeability in hyperglycemic HUVEC model of diabetic retinopathy. Based on evidence from zebrafish and hyperglycemic HUVEC models and diabetic retinopathy patients, we report a hitherto unknown *VEAL2* lncRNA-mediated regulation of PRKCB2, for modulating junctional dynamics and maintenance of endothelial permeability.

**Keywords** diabetic retinopathy; diacylglycerol; endothelial permeability; long non-coding RNA; protein kinase C beta
**Subject Categories** Cell Adhesion, Polarity & Cytoskeleton; RNA Biology; Vascular Biology & Angiogenesis
The EMBO Journal (2021) 40: e107134

## Introduction

Endothelial cells are specialized epithelial cells that form a semi-permeable layer between blood within vessels and surrounding tissues. Endothelial permeability is under the regulation of growth factors such as vascular endothelial growth factor (VEGF) and other chemokines. Pathways activated by these effector molecules result in phosphorylation and internalization of endothelial junctional proteins, altering cell–cell adhesion and thus permeability (Komarova *et al*, 2017). Altered endothelial permeability and dysfunction are observed in a variety of diseases including cardiovascular and neurological disorders, cancer, diabetes, and other metabolic diseases (Cahill & Redmond, 2016; Breier *et al*, 2017; Sweeney *et al*, 2018).

Protein kinase C beta (PRKCB) is a serine–threonine kinase known to be active during endothelial cell proliferation and functions downstream of multiple signaling cascades including VEGF signaling (Simons *et al*, 2016). These cascades lead to the release of secondary messengers such as diacylglycerol (DAG) and $Ca^{2+}$ which act as activators of PRKCB. Activated PRKCB hyperphosphorylates junctional proteins and leads to their translocation and degradation in the cytosol thereby modulating endothelial permeability (Kawakami *et al*, 2002). Human PRKCB gene has two alternatively spliced isoforms, PRKCB1 and PRKCB2. VEGF induces neo-vascularization by the activation of PRCKB2 isoform (Kawakami *et al*, 2002; Simons *et al*, 2016). PRKCB2 activation causes phosphorylation of the tight junction proteins VE-cadherin (CDH5) and beta-catenin (CTNNB1), leading to ubiquitination and proteasomal degradation eventually resulting in increased endothelial permeability in retinal vasculature (Wei *et al*, 2010; Haidari *et al*, 2014). Role of PRKCB2 has been particularly investigated in the context of hyperglycemia and subsequent retinopathy involving excessively permeable eye vasculature

1 CSIR-Institute of Genomics and Integrative Biology, Delhi, India
2 Academy of Scientific and Innovative Research, Ghaziabad, India
3 Laboratory of Calciomics and Systemic Pathophysiology, Regional Center for Biotechnology, Faridabad, India
4 Brien Holden Eye Research Centre, L V Prasad Eye Institute, Hyderabad, India
5 Kannuri Santhamma Centre for Retina and Vitreous, L V Prasad Eye Institute, Hyderabad, India
*Corresponding author. Tel: +91 9650466002; E-mail: vinods@igib.in
**Corresponding author. Tel: +91 9871780662; E-mails: s.sivasubbu@igib.res.in; sridhar@igib.in
†Present address: Department of Computational Biology, Indraprastha Institute of Information Technology, Delhi, India

(Suzuma *et al*, 2002). While the underlying mechanisms of increased vascular permeability and defective neo-angiogenesis in diabetes-related complications are still being explored, the pivotal role of PRKCB2 in regulating endothelial permeability has been well documented in various mammalian model systems (Aiello *et al*, 1997; Durpès *et al*, 2015). PRKCB inhibitors have been tested as therapeutic strategies against vascular anomalies. LY317615 or Enzastaurin is an isozyme-specific PRKCB2 inhibitor explored in diabetic retinopathy (DR). It targets the ATP-binding site of PRKCB2 and inhibits kinase activity. Although a promising candidate in pre-clinical studies, Enzastaurin fell short of efficacy in clinical trials (Bourhill *et al*, 2017). Interestingly, the kinase activity of PRKCB2 has been demonstrated to be regulated by isozyme-specific RNA aptamers (Conrad *et al*, 1994). This highlights the possibility of RNA-based regulation of PRKCB2 *in vivo*.

Long non-coding RNAs (lncRNAs) are non-protein-coding transcripts abundant across eukaryotes with distinct spatio-temporal expression patterns and have surfaced as key regulators in endothelial function. Recent studies have identified lncRNAs that modulate endothelial differentiation and proliferation such as MANTIS, SENCR, GATA6-AS, and LEENE (Boulberdaa *et al*, 2016; Leisegang *et al*, 2017; Miao *et al*, 2018; Neumann *et al*, 2018; Weirick *et al*, 2018). LncRNAs such as MIAT and MALAT1 are in particular known to modulate endothelial permeability (Michalik *et al*, 2014; Yan *et al*, 2015). With the increasing number of emerging lncRNAs, hidden modes of regulation of endothelial permeability at epigenetic, transcriptional, translational, or activity levels are being discovered and provide novel targeting approaches for management of vascular diseases.

In the present study, we show that PRKCB2 is regulated by a novel vascular endothelial-associated lncRNA 2 (*VEAL2*) in both zebrafish and human cells (HUVECs). We provide evidence that the inhibitory role of *VEAL2* on PRKCB2 is enabled through interactions between *VEAL2* and the DAG-binding domain of PRKCB2. We observed decreased human *VEAL2* expression in choroid of diabetic retinopathy (DR) patients and further confirmed its vital role in amelioration of hyperglycemic disease pathophysiology in HUVEC models. Elevation in expression levels of *VEAL2* in blood of patients with diabetic retinopathy poses a potential application as a biomarker for aggravating microvascular complications. This study puts forth *VEAL2* as a potential candidate for lncRNA-mediated inhibition of PRKCB2 in pathological conditions with excessive endothelial permeability.

# Results

## Poly-A RNA sequencing of zebrafish endothelial cells identifies novel vascular endothelial-associated lncRNAs (VEALs)

Using flow cytometry-based approach, endothelial cells (EC) and non-endothelial cells (NEC) were isolated from a double transgenic zebrafish *gib004Tg(fli1a:EGFP;gata1a:DsRed)* at 24–26 h post-fertilization (hpf) (Appendix Fig S1). Poly-A RNA sequencing was performed on both EC and NEC (Appendix Table S1). Using a custom-built bioinformatic pipeline (Fig 1A and described in Appendix Fig S2), we identified 4,897 putative lncRNAs (Dataset EV1). Predicted 4,897 lncRNAs were compared with a comprehensive list of previously identified lncRNAs from ZFLNC database (Hu *et al*, 2018), and 3,866 putative novel zebrafish lncRNAs were identified (Fig 1A). For assessing protein-coding capacity of our lncRNA catalog, translation efficiency scores (TES) were calculated on the basis of ribosomal occupancy on predicted transcripts across eight zebrafish developmental stages. Majority (91%) of the putative novel lncRNAs showed negligible evidence of translation with TES < 0.001, compared to previously documented non-coding RNAs and known protein-coding transcripts (82 and 1.4%, respectively) (Fig 1B). To identify EC-associated lncRNome, expression fold change values were employed and 156 lncRNAs showed 10-fold enrichment in ECs compared to NECs (Fig 1C).

## Characterization of a novel zebrafish vascular endothelial-associated lncRNA-2 (*veal2*)

A candidate multi-exonic lncRNA with highest expression was selected for functional validation and named as zebrafish vascular

---

**Figure 1.  Poly-A RNA sequencing reveals endothelial-associated lncRNome in zebrafish.**

A       Schematic for the experimental workflow and the computational pipeline employed for the discovery and annotation of endothelium-enriched long non-coding RNAs.

B       Distribution of Translation Efficiency Score (TES) across novel lncRNAs identified in this study and RefSeq genes. Box limits indicate the 25th and 75th percentiles as determined by R software; whiskers extend till 5th and 95th percentiles.

C       Differential expression analysis revealed 156 endothelial-enriched lncRNAs with a fold change of at least 10 (closed circles) and 685 lncRNAs at 2-fold (open circles).

D       UCSC browser snapshot of the zebrafish vascular endothelial-associated lncRNA 2 (*veal2*) transcript. 5′ RACE and 3′ RACE data confirmed ends of the *veal2* transcript.

E       Ribosomal pulldown shows lack of occupancy of ribosomes on *veal2*. *fli1a* and *actb* were used as positive controls.

F–I     e-GFP fusion assay confirms lack of peptide formation from *veal2* sequence. (F–G) *mitfa-eGFP* fused transcript. (H–I) *veal2-eGFP* fused transcript. Arrowheads indicating e-GFP expression in *mitfa-eGFP*-injected embryos. Scale bar-100μm.

J       Relative abundance analysis from different subcellular fractions revealed *veal2* is a cytoplasmic lncRNA. Bar graph represents the relative abundance of *veal2* and *actb* transcripts across different fractions of the cell. Data from three different experiments plotted as mean percentage values ± standard deviation.

K       Relative expression of *veal2* across fluorescence-activated cell sorted (FACS) GFP(+) endothelial cells (EC) and GFP(-) non-endothelial cells (NEC). *fli1a* and *actb* were taken as positive control and normalization control, respectively. Data from three different experiments represented as fold change relative to EC values ± standard deviation.

L       Whole-mount in situ expression analysis of the *veal2* transcript across different stages of zebrafish embryos. (LI,II) 1K cell stage, (LIII,IV) 10 hpf stage, and (LV,IV) 28 hpf stage. LI,III,V-Anti-sense-*veal2* probe. (LII,IV,VI) Sense-*veal2* probe. Magnification-2.5X and scale bars-100μm.

M, N    Expression of *veal2* transcript (FPKM scores) across (M) 11 developmental stages and (N) all publically available RNA-seq data of zebrafish's different tissues or cell types compiled by ZFLNC database (Hu *et al*, 2018).

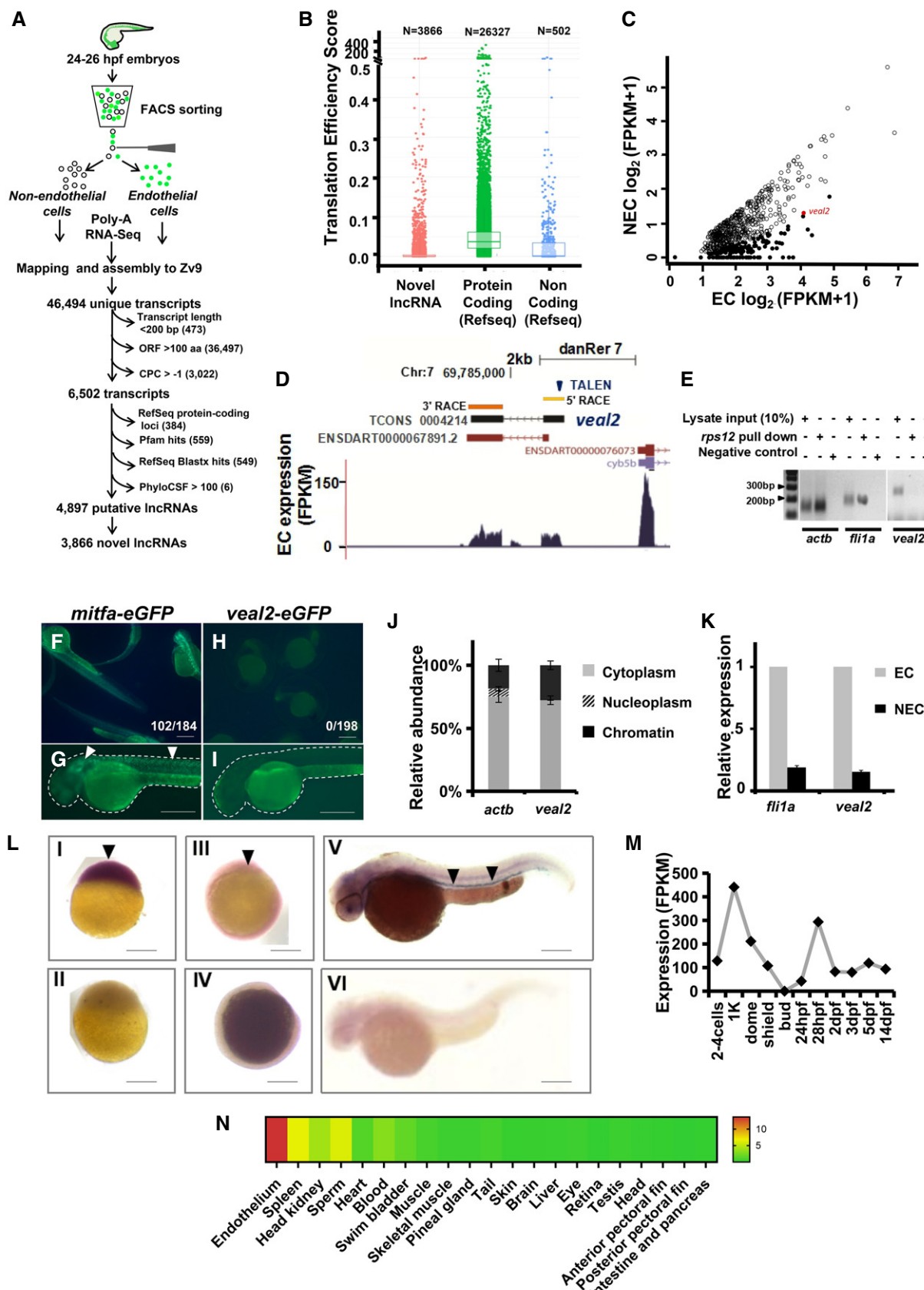

**Figure 1.**

endothelial-associated lncRNA 2 (*veal2*). *veal2* is a 1,127 bp long, bi-exonic gene on zebrafish chromosome 7, proximal to *cytochrome b5 type B (cyb5b)* gene and partially overlapping an lncRNA cataloged in Ensembl database (Ulitsky *et al*, 2011) (Fig 1D, Appendix Fig S3). The full-length *veal2* was confirmed by performing 5' and 3' RACE (Fig 1D, Appendix Fig S4). Longest putative ORF in *veal2* is of 57AA. To confirm the non-protein-coding nature of the predicted *veal2* transcript, ribosomal pulldown assay was performed on zebrafish embryos (Fig 1E). The absence of *veal2* in the ribosomal pulldown fraction confirmed the probable lack of occupancy of ribosomes on transcript under normal cellular conditions. Further fusion RNA product of *e-GFP* in frame with *veal2* was injected in zebrafish, and it confirmed non-protein-coding nature of *veal2* (Fig 1F–I). Subcellular fractionation followed by RT–PCR divulged that *veal2* is enriched in cytosol, apart from being present in chromatin fraction (Fig 1J and Appendix Fig S5). The EC-enriched *veal2* expression was confirmed through RT–PCR (Fig 1K). Expression analysis of *veal2* using *in situ* hybridization and RNA-seq data of developmental stages suggested maternal inheritance of the lncRNA (Fig 1L) followed by modest expression during zebrafish larval development (Fig 1L and M) (Ulitsky *et al*, 2011; Collins *et al*, 2012; Pauli *et al*, 2012). Further tissue-restricted *veal2* expression in vasculature was observed in organogenesis stages (Fig 1LV). Analysis of *veal2* expression in ZFLNC database (Hu *et al*, 2018) and in-house RNA-seq data (Kaushik *et al*, 2013) revealed maximum expression in endothelium followed by spleen and head kidney (Fig 1N). All the data collectively confirmed the non-protein-coding nature of putative novel endothelial specific *veal2*.

## TALEN-mediated editing of *veal2* locus leads to altered vessel integrity in zebrafish

Functional relevance of *veal2* was initially evaluated using splice blocking morpholinos (MOs) targeting *veal2*. Downregulation of *veal2* expression resulted in a significant number of animals with vascular integrity defects (50%, *P*-value-0.5E-4) (Fig EV1A–Q). For complete details, see the Appendix section. For corroborating the role of *veal2* in regulating vascular integrity, we further deleted the entire *veal2* locus using dual transcription activator-like effector nucleases (TALEN) approach, which caused lethality at the larval stages (detailed in Appendix file) (Appendix Fig S7).

Further, we targeted *veal2* using a single pair of TALEN considering possible disruption of RNA secondary structure and hence its function. The TALEN pair targeting 5' end of *veal2* was injected in zebrafish embryos and was confirmed for indels at targeted loci (Fig 2A and B).

We next took a systematic approach to identify and stabilize a *veal2* deletion zebrafish line with a 8-base deletion at positions 26–33 bases of *veal2* transcript and was named as *veal2*[gib005Δ8], as detailed in Appendix section. The cross of *veal2*[gib005Δ8/+] $F_2$ heterozygous animals with background strain *gib004Tg(fli1a:EGFP,-gata1a:DsRed)* (referred to as outcross hereon) gave rise to progeny with vascular integrity defects and cranial hemorrhage in ~ 50% of animals (*P*-value < 1E-4) (Fig 2C and D, and F–O). Randomly, 20 $F_3$ embryos were genotyped, and we found 11 animals were heterozygous for *veal2* allele harboring 8-base deletion (*veal2*-Δ8) (Fig 2C). Zebrafish with the 8-base deletion in *veal2*[gib005Δ8/+] ($F_3$ heterozygous) displayed the disruption in vessel integrity leading to prominent cranial hemorrhage (Fig 2D and F–O) suggesting that haploinsufficiency of *veal2* in heterozygous *veal2*[gib005Δ8/+] animals could cause the hemorrhage phenotype. In order to confirm the role of *veal2* in altering endothelial integrity, a rescue experiment was designed wherein *in vitro*-transcribed full-length *veal2* was exogenously introduced into embryos from the outcross of *veal2*[gib005Δ8/+] animals. About 50% reduction in number of phenotypic animals was achieved by complementing wild-type *veal2* in *veal2*[gib005Δ8/+] embryos (*P*-value = 1E-4) (Fig 2E and P–T). We also demonstrate the absence of any off-target mutation at possible target sites across the zebrafish genome (Appendix Fig S13). The allele-specific expression analysis clearly indicated a ~ 50% reduction in wild-type *veal2* expression in our heterozygous *veal2*[gib005Δ8/+] embryos (Appendix Fig S14A). This validated and confirmed the causal role of *veal2* at RNA level in inducing hemorrhage in *veal2*[gib005Δ8/+] larvae and indicated that *veal2* plays a crucial role in modulating endothelial permeability in zebrafish embryos.

## *veal2* interacts and regulates kinase activity of protein kinase C beta

Next, we decided to determine the possible regulatory molecular mechanism of *veal2* in maintaining endothelial function. We observed that there was no significant change in the expression of

---

**Figure 2.** **TALEN-based gene editing of *veal2* locus caused vascular integrity defects in *gib004Tg(fli1a:EGFP;gata1a:DsRed)* zebrafish embryos.**

A Schematic representation of TALEN design, its injection into one-cell zebrafish embryos, and screening at 2 dpf for any phenotypic changes.

B List of sequences showing indels at the *veal2* locus at somatic level in $F_0$ zebrafish embryos injected with TALEN arms.

C Schematic representation of outcross of *veal2* heterozygous mutant *veal2*[gib005Δ8/+] and the chromatograms representing the two different types of alleles identified on genotyping of 20 embryos randomly.

D Bar graph representing the number of animals which displayed the hemorrhage phenotype across the progeny derived from breeding of two control *gib004Tg(fli1a:EGFP;gata1a:DsRed)* zebrafish and progeny derived from an outcross of *veal2*[gib005Δ8/+] zebrafish. Data from three different experiments plotted as mean percentage values ± standard deviation.

E Bar graph representing the number of animals which displayed the hemorrhage phenotype across the progeny derived from an outcross of *veal2*[gib005Δ8/+] zebrafish injected with *veal2* RNA and vehicle control separately. Data from three different experiments plotted as mean percentage values ± standard deviation.

F–T Representative images of 2 dpf zebrafish which displayed the rescue of the vascular integrity defects in the progeny of an outcross of *veal2*[gib005Δ8/+] zebrafish upon complementing with the wild-type (WT) *veal2* RNA. (F–J) Control zebrafish embryos. (K–O) *veal2*[gib005Δ8/+] embryos injected with vehicle control. (P–T) *veal2*[gib005Δ8/+] embryos complemented with *veal2* RNA. (F, K, P) Bright field. (G, L, Q) mRFP. (H, M, R) Animals stained with O-dianisidine stain. (I, N, S) eGFP. (J, O, T) Merged eGFP and mRFP filters. Arrowheads show the presence of hemorrhage due to the vascular integrity defects. (F–H, K–M, P–R) Magnification-5× and scale bar-100 μm. (I–J, N–O, S–T) Magnification-20× and scale bar-50 μm.

Data information: All the experiments N ≥ 3. ****P*-value < 1E-4 and ***P*-value = 1E-4. Statistics-unpaired two-tailed t-tests.
Source data are available online for this figure.

---

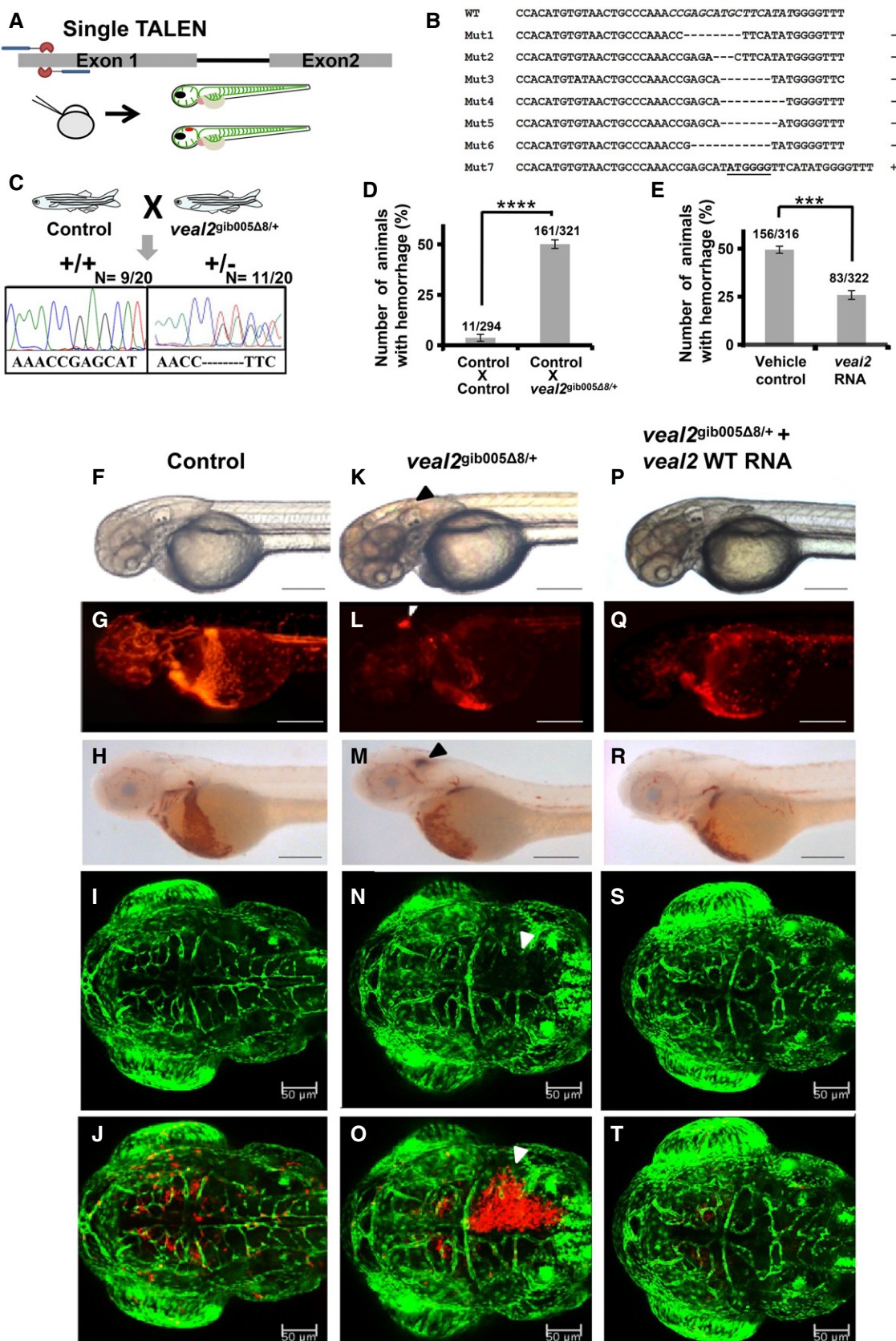

**Figure 2.**

all five (*cyb5b, dhx38, zgc:162592, kars,* and *rpl13*) neighboring genes in 600 kb proximity of *veal2* loci in *veal2*[gib005Δ8/+] animals, indicating lack of its cis-activity (Appendix Table S4). A shift in the electrophoretic migration of wild-type full-length *veal2* was observed when incubated with protein lysate of zebrafish embryos, suggesting potential interaction between lncRNA and proteins (Appendix Fig S14B). To identify specific interacting proteins, RNA anti-sense pulldown (RAP) of *veal2* was performed followed by mass spectrometry (Fig 3A, Appendix Table S3). We selected four candidate proteins (results in Appendix file) and tested their interaction with *veal2* by RAP followed by immunoblotting, which confirmed potential interaction between *veal2* and proteins Ephb3 and Prkcbb, independently (Fig 3B). As a complementary approach, RNA immunoprecipitation (RIP) was performed to capture Ephb3 and Prkcbb and detect the enrichment of *veal2* (Fig 3C). Relative abundance of *veal2* was ~ 10-fold (*P*-value = 1E-3) higher in Prkcbb-IP in comparison with IgG-IP (Fig 3D). The results taken together indicated that there is a direct interaction between Prkcbb protein and *veal2* in zebrafish.

Prkcbb is a serine–threonine kinase conserved across vertebrates (~ 70%) and has ~ 86% protein sequence similarity between zebrafish and humans (Appendix Fig S16A). Considering this high sequence conservation, we tested the effect of zebrafish *veal2* RNA on human PRKCB2 protein using an *in vitro* kinase assay. Addition of 25 nM of full-length wild-type *veal2* to the kinase assay reduced PRKCB2 activity to ~ 45% (*P*-value < 1E-3) (Fig 3E). Full-length *veal2*-anti-sense (*veal2*-AS) and *veal2* lacking 8 bp (*veal2*-Δ8) did not show any effect on PRKCB2 activity (Fig 3E). This suggested that wild-type *veal2* had the ability to inhibit human PRKCB2 kinase activity. We further observed that full-length *veal2*, and not shorter fragments, was necessary to inhibit PRKCB2 activity (Appendix Fig S17B).

To confirm the specific inhibitory effect of *veal2* on PRKCB2, we performed small molecule-based rescue in *veal2*[gib005Δ8/+] animals using Enzastaurin. Enzastaurin is a chemical molecule known to block the ATP-binding catalytic domain of human PRKCB2. Enzastaurin treatment rescued hemorrhage phenotype in about 52% of offspring from the outcross of *veal2*[gib005Δ8/+] animals compared to DMSO treatment (*P*-value < 1E-4) (Fig 3F–K and L). This

complementation of *veal2* function by Enzastaurin in *veal2*[gib005Δ8/+] animals confirmed that *veal2* negatively regulates zebrafish Prkcbb activity.

### *veal2* competes with DAG to regulate Prkcbb activity

To understand the interaction of Prkcbb with *veal2* at molecular resolution, we employed a combinatorial *in silico* approach of molecular dynamic simulations followed by docking using HDOCK (Appendix Figs S18 and S19A and B). We identified four motifs within *veal2* interacting with Prkcbb protein, spanning the following RNA base positions (5'-3'): motif-1(549–560), motif-2(584–587), motif-3(678–680), and motif-4(844–846) (Fig 4A, Appendix Fig S20A and C). Variants of *veal2* RNA lacking each of the motifs independently were injected separately into *veal2*[gib005Δ8/+] animals. In the complementation assay, *veal2* RNA that harbored deletion of motif-1, motif-2, or motif-4 could successfully complement the *veal2*[gib005Δ8/+] animals at varying degrees (Fig 4C). However, *veal2* variant lacking motif-3(678–680) failed to rescue the cranial hemorrhage phenotype (Fig 4C), suggesting the potential functionality of motif-3 in regulating Prkcbb activity. This was further supported by *in vitro* kinase assay: Only deletion of motif-3 led to perturbation in inhibitory function of *veal2* on Prkcbb's kinase activity (Appendix Fig S20C). The bases in the *veal2* motif-3 were predicted to form hydrogen bonds with H116 and S119 in the C1 domain of Prkcbb protein. We termed motif-3 of *veal2* as C1 domain binding lncRNA motif (CLM) (Fig 4D). The CLM-interacting residues lie in the conserved hydrophobic stretch along the C1 domain and directly interact with lipid activators of the kinase, as shown in an earlier study using rat PRKCB2 (Leonard *et al*, 2011) (Appendix Fig S16B and S20D). Diacylglycerol (DAG), a lipid secondary messenger downstream of VEGF pathway, binds to C1 domain (also known as allosteric site) of PRKCB2 leading to its activation. Thus, the proximity of *veal2*'s CLM and the predicted hydrophobic stretch suggests that *veal2* interacts with DAG-binding domain in the Prkcbb protein and regulates its activity. Further, we predicted the implication of the 8-base deletion identified in *veal2*[gib005Δ8/+] animals on the interaction with Prkcbb. The HDOCK-based prediction determined that the 8-base deletion alters the RNA folding, preventing the

**Figure 3.** *veal2* lncRNA interacts and negatively regulates Prkcbb protein in zebrafish.

A Schematic of the methodology adopted for the identification of protein interacting partners of *veal2* using a RAP-MS-based approach.

B RAP-MS followed by Western blotting validated Prkcbb and Ephb3 as interacting protein partners with *veal2*.

C RNA immunoprecipitation of *veal2* was performed by pulldown of Prkcbb and Ephb3, followed by qRT–PCR. IgG pulldown was performed as control. Gel image shows the amplification of *veal2* in different RNA immunoprecipitation samples, along with 10% input control, as marked by arrowheads.

D qRT–PCR-based quantification of *veal2* transcript across Prkcbb, Ephb3, and IgG immunoprecipitations. *actb* was used as normalization control. Data from three different biological replicates represented as mean fold change values ± standard deviation.

E Relative kinase activity of human PRKCB2 under standard conditions and in the presence of various concentrations of the WT *veal2* RNA, *veal2*-Δ8 RNA, and *veal2*-AS RNA. 52 nM of the PRKCB2 protein was used per reaction. Data from three different experiments plotted as mean fold change values ± standard deviation.

F–K Enzastaurin treatment rescues hemorrhage phenotype in *veal2*[gib005Δ8/+] zebrafish embryos indicated by rescue of the vascular integrity defects in *veal2*[gib005Δ8/+]. Arrowheads show the presence of hemorrhage due to the vascular integrity defects. Experiment was repeated in biological replicates, and a total no. of embryos scored are mentioned in figure. (F–I) Magnification-5× and scale bar-100 μm. (J–K) Magnification-20× and scale bars-50μm.

L Relative number of animals that displayed the hemorrhage phenotype across the progeny of the outcross of *veal2*[gib005Δ8/+] zebrafish. Number of phenotypic animals upon treatment with Enzastaurin was normalized with the number of phenotypic animals when treated with DMSO. Data from three different experiments plotted as mean fold change values ± standard deviation.

Data information: All the experiments N ≥ 3. ***P-value < 1E-3 and ****P-value < 1E-4. Statistics: (D,L) unpaired two-tailed t-tests. (E) Two-way ANOVA with Bonferroni's multiple data comparison.

Source data are available online for this figure.

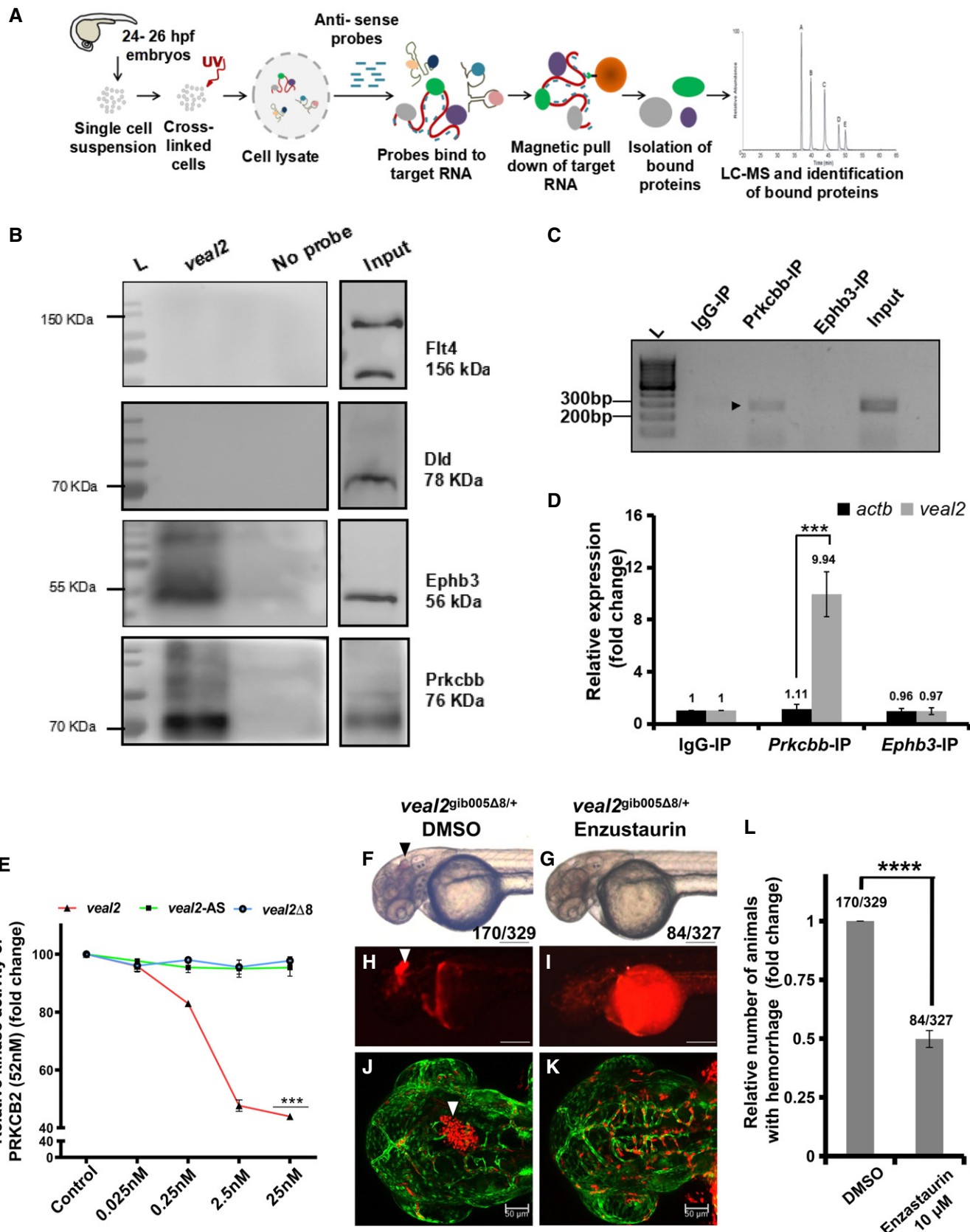

**Figure 3.**

interaction between *veal2*'s CLM and Prkcbb DAG-binding domain (Fig 4B, Appendix Fig S20B). To further evaluate the nature of inhibition by *veal2* in the presence of DAG, PRKCB2 kinase assay was performed using varying concentrations of both *veal2* and DAG. The *veal2* concentration dose was used between 0.025–25 nM based on previous experiment (Fig 4E). In the absence of *veal2*, PRKCB2 activity followed a hyperbolic pattern with increasing concentrations of DAG. In the presence of 0.025 and 0.25 nM of *veal2*, minor shifts in the hyperbolic curves were observed. In the presence of 2.5 and 25 nM of *veal2*, the PRKCB2 activity maxima was lowered at lesser concentrations of DAG, which however approached the maximum activity with higher concentrations of DAG (Fig 4E). The initial shift in the $K_m$ values followed by reactions approaching similar $V_{max}$ values indicated that *veal2* and DAG compete for binding at allosteric site of PRKCB2 protein (Fig 4E).

In order to study the potential role of *veal2* in regulating DAG-mediated activation and membrane translocation of Prkcbb for controlling endothelial cell permeability, we evaluated the subcellular localization of Prkcbb in *veal2*[gib005Δ8/+] animals. We isolated the membrane and cytoplasmic fractions of the cells and performed immunoblotting of Prkcbb. Prkcbb levels were compared among the two fractions from whole embryo lysates of control and *veal2*[gib005Δ8/+] animals. Total Prkcbb levels were equal in whole animal lysates (Fig 4G); however, Prkcbb was significantly enriched in the membrane fraction than in cytoplasmic fraction in the *veal2*[gib005Δ8/+] animals, compared to the control (Fig 4F and G). This indicated the uncontrolled activation and excessive plasma membrane translocation of Prkcbb protein in *veal2*[gib005Δ8/+] animals lacking functional *veal2*.

## Immunoprecipitation of PRKCB in HUVECs identifies a human ortholog of *veal2*

The possibility of a human ortholog of *veal2* was hinted at by the interaction and inhibitory activity of *veal2* on human PRKCB2 protein. By nucleotide sequence similarity we could not identify a conserved ortholog of *veal2* in humans. We then attempted an experimental approach to identify the human ortholog of *veal2* by capturing PRKCB-associated RNAs in HUVECs by RIP-seq (Fig 5A). The RIP-seq of PRKCB in HUVECs showed enrichment of 20 transcripts (2 biological replicates), of which 6 transcripts were lncRNAs

and only one candidate-AC008440.2 was significantly enriched (*P*-value-5E-5) with respect to IgG pulldown (Fig 5B and Dataset EV4). This human bi-exonic anti-sense lncRNA is 423 bases long, arises from the 3' UTR of *myeloid-associated differentiation marker (MYADM)* gene and shows no sequence conservation with *veal2*. We named this novel lncRNA as human *VEAL2* (*VEAL2*) (Fig 5C).

*VEAL2* is a legitimate lncRNA according to CPAT (Wang *et al*, 2013) (coding probability 0.0008), PhyloCSF (Lin *et al*, 2011) and CPC (Kong *et al*, 2007) (coding potential score −0.528702). FANTOM CAGE-seq data, 3' RACE (Fig 5C) and RIP-sequencing data identified *VEAL2* as an independent transcript. Absolute quantification of *VEAL2* showed approximately 0.024 copies per HUVEC cell (Fig 5D), which translates to ~ 562 copies of *VEAL2* per μg of HUVEC RNA. The expression analysis of *VEAL2* in different primary cell lines revealed an enriched expression of *VEAL2* in HUVECs (Appendix Fig S21B). Although *VEAL2* has a putative ORF of 25AA but to determine the coding potential *in vivo*, we injected *VEAL2*-e-GFP fusion transcript in zebrafish embryos. Lack of GFP expression in *VEAL2*-e-GFP fusion transcript-injected zebrafish embryos confirmed non-protein-coding nature of human *VEAL2* (Fig 5E–H).

We complemented human *VEAL2* in *veal2*[gib005Δ8/+] embryos and successfully rescued the hemorrhage phenotype (Fig 5I–N), with a 45% reduction in the number of phenotypic animals (*P*-value < 1E-3) (Appendix Fig S21C). This indicated that human *VEAL2* is a functional ortholog of zebrafish *veal2*. Single molecule FISH assay for *VEAL2* in HUVECs showed cytoplasmic localization (Fig 5O–Q) as observed for *veal2* in zebrafish (see Fig 1J). Further, we were keen to know the *in vivo* interaction of PRKCB and *VEAL2*. Co-IF and FISH in HUVECs displayed significant colocalization of signal of *VEAL2* with PRKCB (~ 70%) (*P*-value < 1E-4) (Fig 5R–U). Although *veal2* and *VEAL2* showed no obvious nucleotide sequence and structure similarity, *in silico* simulation-based studies demonstrated that *VEAL2* has an affinity to form hydrogen bonds with PRKCB2 protein at AA position 123 and we called it CLM (motif-3 in zebrafish). Interestingly, this residue is part of DAG-binding domain of PRKCB2 protein (Leonard *et al*, 2011) (Fig EV2A–C) suggesting that similar to *veal2*, *VEAL2* could also interfere in DAG and PRKCB2 interaction. We also tested the effect of *VEAL2* on human PRKCB2 protein using *in vitro* kinase assay. Addition of 140 nM of the full-length wild-type *VEAL2* to the kinase reaction reduced the PRKCB2 activity to ~ 55% (*P*-value < 1E-3) (Fig 5V). *VEAL2* also displayed

---

**Figure 4. *veal2* interacts with DAG-binding C1 domain of Prkcbb protein and regulates kinase activation.**

A, B   *In silico* docking of *veal2* and Prkcbb identified putative 4 interaction motifs in *veal2*. Motif-1(549, 551-553, 559, 560) in purple, motif-2(584-587) in orange, motif-3 (678–680) in red, and motif-4(844–846) in green. The 8bp deletion in *veal2*-Δ8 RNA led to change in folding and altered interaction with Prkcbb.

C   Scatter plot showing the number of animals which displayed the hemorrhage phenotype across the progeny derived from an outcross of *veal2*[gib005Δ8/+] zebrafish injected with IVTs of different *veal2* variants. Control indicates *veal2*[gib005Δ8/+] zebrafish without complementation. Data from three different experiments plotted as individual values; the middle bar represents mean percentage, and the error bar represents ± standard deviation.

D   The site of interaction of the motif-3 of *veal2* (CLM) with the Prkcbb protein indicates that it lies in a previously known DAG-binding site. Both RNA (pink) and protein (purple) structures are shown as ribbon models. The distances between nucleotides of motif-3 of *veal2* and the 2 amino acids of Prkcbb are given in Å units.

E   Relative kinase activity of human PRKCB2 with various concentrations of DAG without or with different concentrations of *veal2* WT RNA (0, 0.025, 0.25, 2.5, 25 nM). PRKCB2 activity without DAG was used for normalization. Data from three different experiments plotted as mean fold change values ± standard deviation.

F   Abundance of Prkcbb protein in total cell (T), cytoplasmic (C), and membrane (M) fractions of cells from 2 dpf *gib004Tg(fli1a:EGFP;gata1a:DsRed)* and *veal2*[gib005Δ8/+] zebrafish embryos. Arrowhead indicates Prkcbb enrichment in the membrane fraction of *veal2*[gib005Δ8/+] zebrafish embryos.

G   Relative quantification of Prkcbb localization in cytoplasmic and membrane fractions of control and *veal2*[gib005Δ8/+] embryos. Data from three different experiments plotted as mean percentage values ± standard deviation.

Data information: All the experiments N ≥ 3.
Source data are available online for this figure.

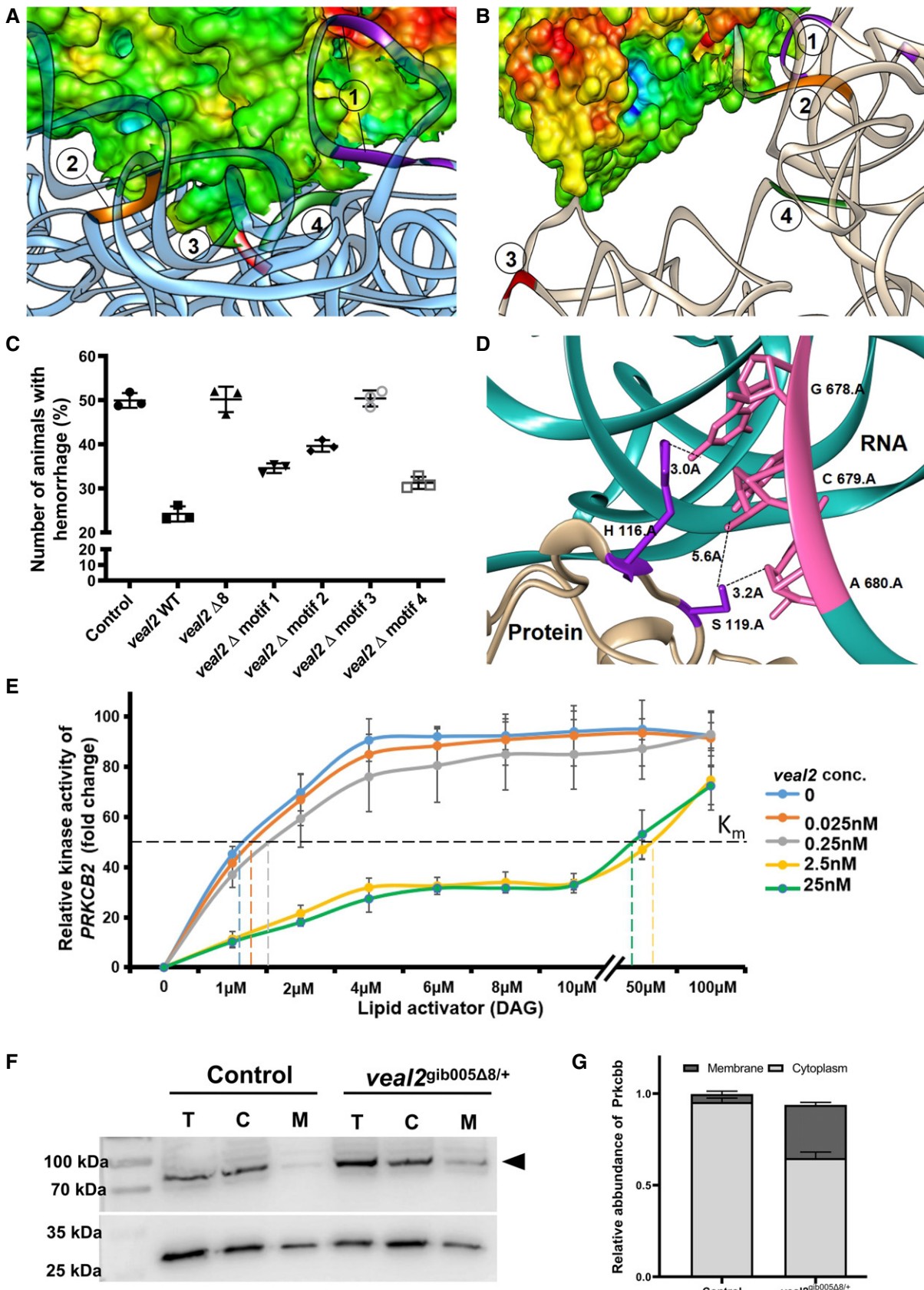

**Figure 4.**

an inhibitory effect on the kinase activity in *in vitro* conditions. Further deletion of the CLM bases in *VEAL2* (*VEAL2-ΔCLM*) resulted in perturbation of the inhibitory function of *VEAL2* on PRKCB2's kinase activity (Fig EV2D).

### *VEAL2* regulates permeability and angiogenesis in HUVECs by modulating junctional dynamics controlled by PRKCB

We examined the functional significance of *VEAL2* in human ECs by performing plasmid-based overexpression and siRNA-mediated knockdown in HUVECs. *VEAL2* levels in plasmid-based overexpression and siRNA-induced knockdown in HUVEC models were found to be significantly altered as tested by RT–PCR and FISH, respectively (Figs 6A and F, and EV3E and F). The effect of *VEAL2* on the angiogenic property of HUVEC was assayed. Both overexpression and knockdown of *VEAL2* in HUVECs showed formation of colonies of cells on Matrigel but had significantly reduced tubulogenesis and junctions between tubes compared to controls (*P*-value < 1E-4) (Fig 6B–D and G–I, respectively). The absence of any deviation in lactate dehydrogenase (LDH) levels confirmed that the anti-angiogenesis effect is not due to cell toxicity (Appendix Fig S22D). In a scratch-wound assay, *VEAL2* overexpression displayed enhanced proliferation and migration of HUVECs and downregulation of *VEAL2* showed reduced wound repair (Fig EV3A–D, respectively).

We evaluated the role of *VEAL2* in regulating endothelial permeability. Efflux of dextran-FITC across the HUVEC monolayer was significantly less (*P*-value < 1E-3) in *VEAL2*-overexpressed cells (Fig 6E). In contrast, *VEAL2* knockdown cells displayed significantly high dextran-FITC efflux (*P*-value < 1E-3), indicating *VEAL2*-mediated regulation of permeability (Fig 6J). Further to demonstrate the regulation of PRKCB2 by *VEAL2* in maintaining junctional dynamics, the subcellular expression of VE-cadherin (CDH5), beta-catenin (CTNNB1), and PRKCB was detected using immunofluorescence. In the control plasmid or control siRNA

treated cells, PRKCB was distributed in the cytoplasm and membrane (Fig 6K and M). In *VEAL2*-overexpressed cells, signals of PRKCB were observed mostly in cytoplasm only (Fig 6L and W). Upon knockdown of *VEAL2*, localization of PRKCB increased on the membrane significantly (Fig 6N and X). This was supporting the pattern noticed in *veal2*^gib005Δ8/+ animals (Fig 4F and G), indicating the role of *VEAL2* in regulating PRKCB activity and translocation. The junctional proteins CDH5 and CTNNB1, which are substrates of PRKCB phosphorylation, showed excessive localization on the membrane in *VEAL2*-overexpressed cells (Fig 6O, P and Y; S, T and AA, respectively). In contrast, knockdown of *VEAL2* displayed significant diffused expression of CDH5 and CTNNB1 in cytoplasm compared to control (Fig 6Q, R and Z; U, V and AB, respectively). We tested the effect of *veal2* on angiogenesis and permeability in HUVECs which corroborated similar effects as seen with *VEAL2* indicating their functional conservation (Appendix Fig S22). In summary, *VEAL2* plays a pivotal role in regulating junctional dynamics controlled by PRKCB to maintain endothelial permeability. Probing of *in-cell* endogenous kinase activity of PRKCB2 upon overexpression of the various *VEAL2* variants and during *VEAL2* knockdown conditions clearly establishes the *in vivo* regulatory role of *VEAL2* on the kinase activity of PRKCB2 (Fig EV4A and B, respectively).

### *VEAL2* is associated with microvascular complications in human diabetic retinopathy

Under diabetic conditions, DAG levels increase and PRKCB2 is hyperactivated, leading to hyperpermeability of retinal vessels and edema, resulting in a clinical conditions namely diabetic retinopathy (DR) (Suzuma *et al*, 2002; Bourhill *et al*, 2017). To obtain hints toward *VEAL2*-mediated regulation of PRKCB2 in DR, we estimated *VEAL2* levels in human choroid tissue samples from donor eyes of type II diabetes mellitus (DM) patients with early retinal vasculature changes. Early symptoms of retinopathy in these DM eye samples

---

**Figure 5. A conserved human *VEAL2* (*VEAL2*) interacts and regulates human protein kinase c beta protein.**

A   Schematic of the methodology adopted for the identification of RNA interacting partners of PRKCB in HUVECs using a RIP-seq-based approach.

B   RIP-seq of PRKCB in HUVECs identified a candidate lncRNA with significant q-value (0.002) and high expression.

C   Representation of genomic location of AC008440.2-human vascular endothelial-associated lncRNA 2 (*VEAL2*) transcript. It is anti-sense to 3′ UTR of known protein-coding *myeloid-associated differentiation marker (MYADM)* gene. 3′RACE analysis confirmed *VEAL2* as an independent transcript. Reads from PRKCB-RIP-seq and IgG-RIP-seq mapping to *VEAL2* loci are also given. Blue highlights reads mapping to +ve strand, and red highlights reads mapping to -ve strand.

D   Absolute quantification of *VEAL2* in HUVECs revealed 0.024 copies per cell. $10^2$ to $10^8$ copies of *VEAL2* RNA were used to make the standard curve. Data from three different experiments plotted as mean values $\pm$ standard deviation.

E–H   e-GFP fusion assay confirms lack of peptide formation from *veal2* sequence. (E–F) *mitfa-eGFP* fused transcript. (G–H) *VEAL2-eGFP* fused transcript. Arrowheads indicating e-GFP expression in *mitfa-eGFP*-injected embryos. Scale bar-100 μm.

I–N   Complementation of *VEAL2* in *veal2*^gib005Δ8/+ embryos significantly rescued hemorrhage phenotype. Arrowheads show the presence of hemorrhage due to the vascular integrity defects. (I–L) Magnification-5× and scale bar-100 μm. (M–N) Magnification-20× and scale bars-50 μm.

O–Q   Single molecule FISH (smFISH) of *VEAL2* in HUVECs shows its cytoplasmic localization. (O) *VEAL2* in CAL Fluor Red (610 nM). (P) DAPI. (Q) Merged image for *VEAL2* and DAPI. Magnification-100× and scale bar-5 μm.

R–T   Co-IF for PRKCB and smFISH of *VEAL2* highlight their colocalization. (R) PRKCB in the GFP channel. (S) *VEAL2* in CAL Fluor Red (610 nM). (T) Merged image for PRKCB and *VEAL2*. Magnification-100× and scale bar-20 μm.

U   Bar graph represents the colocalization rate (%) of *VEAL2* with PRKCB and CAMKIID proteins in HUVECs. Data from three different experiments plotted as individual values with mean percentage values $\pm$ standard deviation.

V   Relative kinase activity of human PRKCB2 under standard conditions and in the presence of various concentrations of the WT *VEAL2* RNA and *VEAL2*-AS RNA. 52 nM of the PRKCB2 protein was used per reaction. Data from three different experiments plotted as mean fold change values $\pm$ standard deviation.

Data information: All the experiments $N \geq 3$. ***$P$-value < 1E-3. Statistics: (U) unpaired two-tailed t-test. (V) Two-way ANOVA with Bonferroni's multiple data comparison.
Source data are available online for this figure.

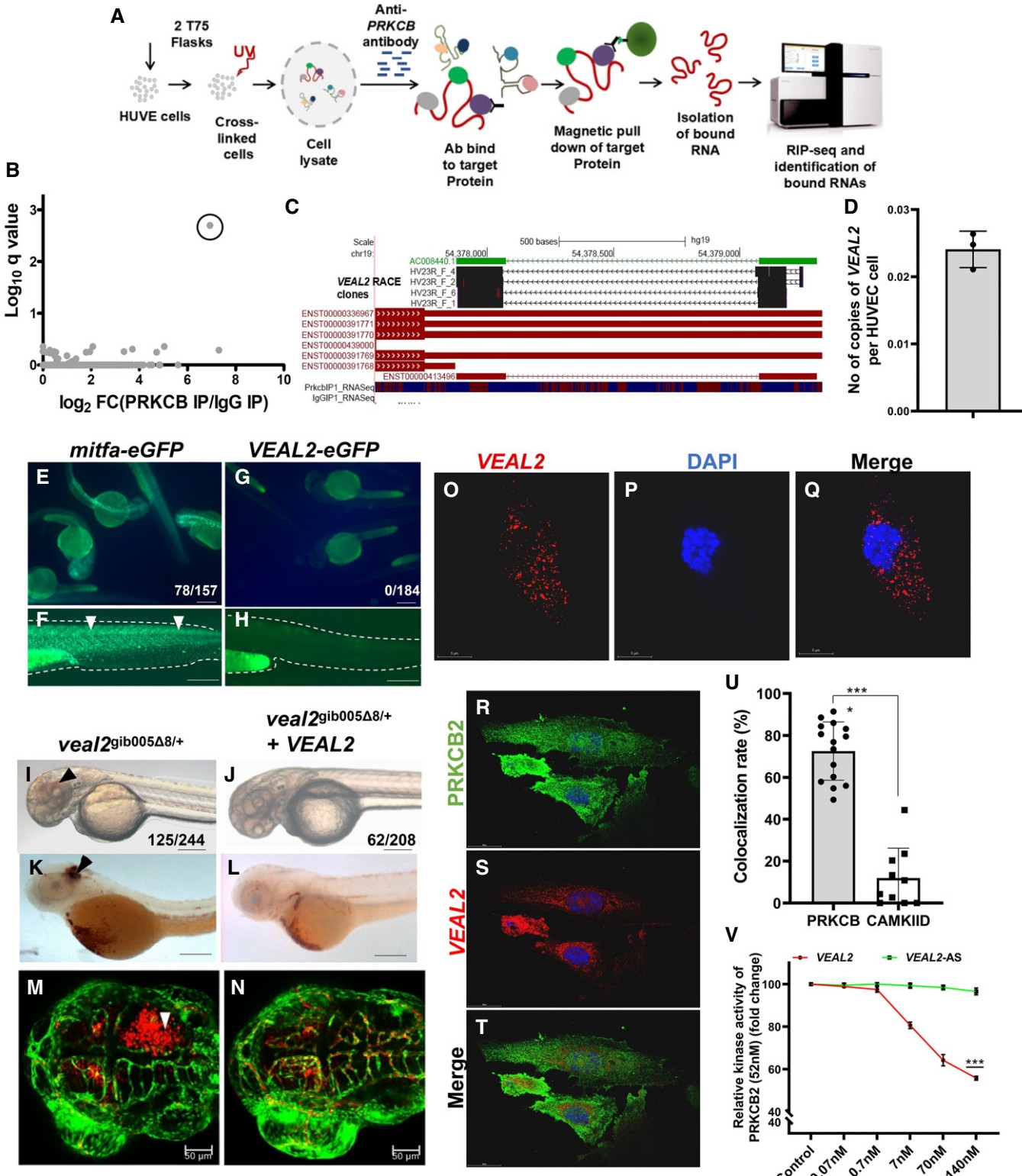

Figure 5.

were confirmed by immunohistochemistry, which exhibited degeneration of the ganglion layer, microaneurysm, arteriolar dilatation, mild edema, and hemorrhage (Fig EV5A–F). *VEAL2* expression was

significantly low (*P*-value < 1E-4) in the choroid tissue of DR patients (*n* = 8) compared to the patients without DR (*n* = 8) (Fig 7A). We checked the *VEAL2* expression in the blood samples of DM

($n$ = 50), non-proliferative DR (NPDR) ($n$ = 50), and proliferative DR (PDR) patients ($n$ = 50) and compared them with *VEAL2* levels in blood samples of control individuals ($n$ = 50). Surprisingly, there was a significant increase in *VEAL2* expression with increase in vascular remodeling defects in diabetic patients from DM to PDR ($P$-value < 1E-3) (Fig 7B). ROC curve analysis from normalized Ct values showed significant correlation of increased *VEAL2* levels in PDR patients with severe vascular defects, as area under the curve (AUC) was 0.766 for *VEAL2* (95% CI: 0.674 to 0.857, $P$ = 0.0001) (Fig 7C). In order to understand the influence of *VEAL2* on neo-angiogenesis, we assessed diabetes-induced neo-vascularization in the normal avascular fibrous membrane of the retina. We observed increased *VEAL2* levels by ~ 34-fold ($P$-value < 1E-4) in vascularized fibrous membranes of DR patients ($n$ = 7) compared to avascular fibrous membranes of the controls ($n$ = 7) (Fig EV5G). Taken together with the results, differential *VEAL2* expression under diabetic conditions suggests an association of *VEAL2* with DR pathogenesis.

To further confirm and identify the potential therapeutic utility of *VEAL2* in DR condition, we tested whether *VEAL2* overexpression in the hyperglycemic HUVEC model could compensate for pathophysiological defects. We created a hyperglycemic HUVEC model of DR by growing them at high glucose concentration and observed an increase in the efflux of dextran-FITC across HUVEC junctions by ~ 165%, indicating enhanced permeability in response to glucose exposure (Fig 7D). Expression levels of *VEAL2* in hyperglycemic HUVEC model were detected significantly low (Fig EV5H). Expression of CDH5, CTNNB1, and PRKCB was checked in the hyperglycemia disease model, and we observed that PRKCB was localized

extensively in the membrane, while the signals of CDH5 and CTNNB1 were diffused and not exclusively on the membrane (Fig 7E, F and Q; I, J and R; M, N and S). After establishing the hyperglycemia model in HUVECs, the potential of *VEAL2* to modulate endothelial permeability was checked in this system. Efflux of dextran-FITC in *VEAL2*-overexpressed HUVECs was ~ 120% indicating partial rescue of hyperpermeability in comparison with ~ 165% in control glucose-treated HUVECs (Fig 7D). Immunofluorescence signals showed CDH5 and CTNNB1 were retained on the membrane upon overexpression of *veal2* or *VEAL2* (Fig 7G, H and Q; K, L and R). PRKCB showed predominantly restricted expression in cytoplasm upon overexpression of *veal2* or *VEAL2* (Fig 7O, P and S). Taken together, this indicated that *VEAL2* can modulate PRKCB activity and restrict translocation of PRKCB to the cell membrane in hyperglycemia, mitigating subsequent turnover of junctional complexes and maintaining endothelial cell integrity.

## Discussion

In the current study, cell-type-specific transcriptomics was performed to identify endothelium-enriched lncRNAs from developing zebrafish embryos. We further investigated the role of a novel candidate lncRNA *veal2* in vascular development and maintaining endothelial junction integrity. Zebrafish harboring an 8-base deletion in *veal2* displayed hemorrhage, depicting the characteristic feature of endothelial junction integrity defects. LncRNA *veal2* directly interacts and regulates the kinase activity of Prkcbb in zebrafish. Treatment of *veal2*$^{gib005\Delta8/+}$ zebrafish embryos with a

**Figure 6.  *VEAL2* regulates permeability and angiogenesis in HUVECs by controlling junctional dynamics.**

A   *VEAL2* RNA levels significantly increased upon overexpression of *VEAL2* plasmid compared to control. Bar graph representing relative expression of *VEAL2* in control pcDNA3.1 plasmid (1 μg) and *VEAL2* in pcDNA3.1 plasmid (1 μg) for overexpression in HUVECs. GAPDH was taken as normalization control. Data are acquired from 3 different biological replicates and shown as mean fold change values ± standard deviation.

B–D   Overexpression of *VEAL2* in HUVECs displayed massive reduction in tube formation in Matrigel compared to control. (B, C) Magnification-5× and scale bar-50 μm. (D) Dot plot representing quantification of number of junctions formed between the vessels in control cells and *VEAL2*-overexpressed cells grown on Matrigel. Data from different fields of 4 different technical replicates of 1 biological replicate are represented. Data are shown as individual values; the middle bar represents the mean, and the error bar represents ± standard deviation.

E   Overexpression of *VEAL2* significantly changes efflux of dextran-conjugated FITC measuring permeability levels. Bar graph representing relative quantification of efflux of dextran-conjugated FITC measuring permeability levels in control and *VEAL2*-overexpressed HUVECs. Data obtained from 3 different biological replicates and plotted as mean percentage fold change values ± standard deviation.

F   siRNA-mediated knockdown of *VEAL2* significantly reduces expression of *VEAL2* in HUVECs. Bar graph representing relative expression of *VEAL2* in control siRNA and *VEAL2* targeting siRNA-transfected HUVECs. Data are acquired from 3 different biological replicates and shown as mean fold change values ± standard deviation.

G–I   Knockdown of *VEAL2* significantly reduced tube formation in Matrigel. (G–H) Magnification-5× and scale bar-50 μm. (I) Dot plot representing quantification of number of junctions formed between the vessels in control siRNA and *VEAL2* targeting siRNA-treated HUVECs. The HUVECs were grown on Matrigel. Data from different fields of 4 different technical replicates of 1 biological replicate are represented. Data are shown as individual values; the middle bar represents the mean, and the error bar represents ± standard deviation.

J   siRNA-mediated knockdown of *VEAL2* significantly changes efflux of dextran-conjugated FITC measuring permeability levels. Bar graph representing relative quantification of efflux of dextran conjugated FITC for measuring permeability levels in control siRNA- and *VEAL2* siRNA-transfected HUVECs. Data obtained from 3 different biological replicates and plotted as mean percentage fold change values ± standard deviation.

K–AB   *VEAL2* regulates junctional dynamics by interacting with PRKCB. Overexpression of *VEAL2* retains PRKCB mostly in cytoplasm and keeps strong junctional assembly formation of CDH5 and CTNNB1 on the membrane. Knockdown of *VEAL2* led to migration of PRKCB on membrane and henceforth degradation of junctional assembly of CDH5 and CTNNB1. (K–N, W–Z) PRKCB. (O–R, Y–Z) CDH5. (S–V, AA–AB) CTNNB1. (K–V) Magnification-60× and scale bar-15 μm. Arrowheads indicate representation of signals of proteins in HUVECs. (W–AB) Dot plot representing quantification of protein signal localization in membrane/total fraction. The quantification was done using ImageJ. Data from cells of different fields of 3 technical replicates of 1 biological replicate are presented as representation. Data are shown as individual values; the middle bar represents the mean, and the error bar represents ± standard deviation.

Data information: All the experiments $N \geq 3$. (D, I, W–AB) Data from cells of different fields of 1 biological replicate are presented as representation. **$P$-value < 1E-2, ***$P$-value < 1E-3, and ****$P$-value < 1E-4. Statistics-unpaired two-tailed t-test.
Source data are available online for this figure.

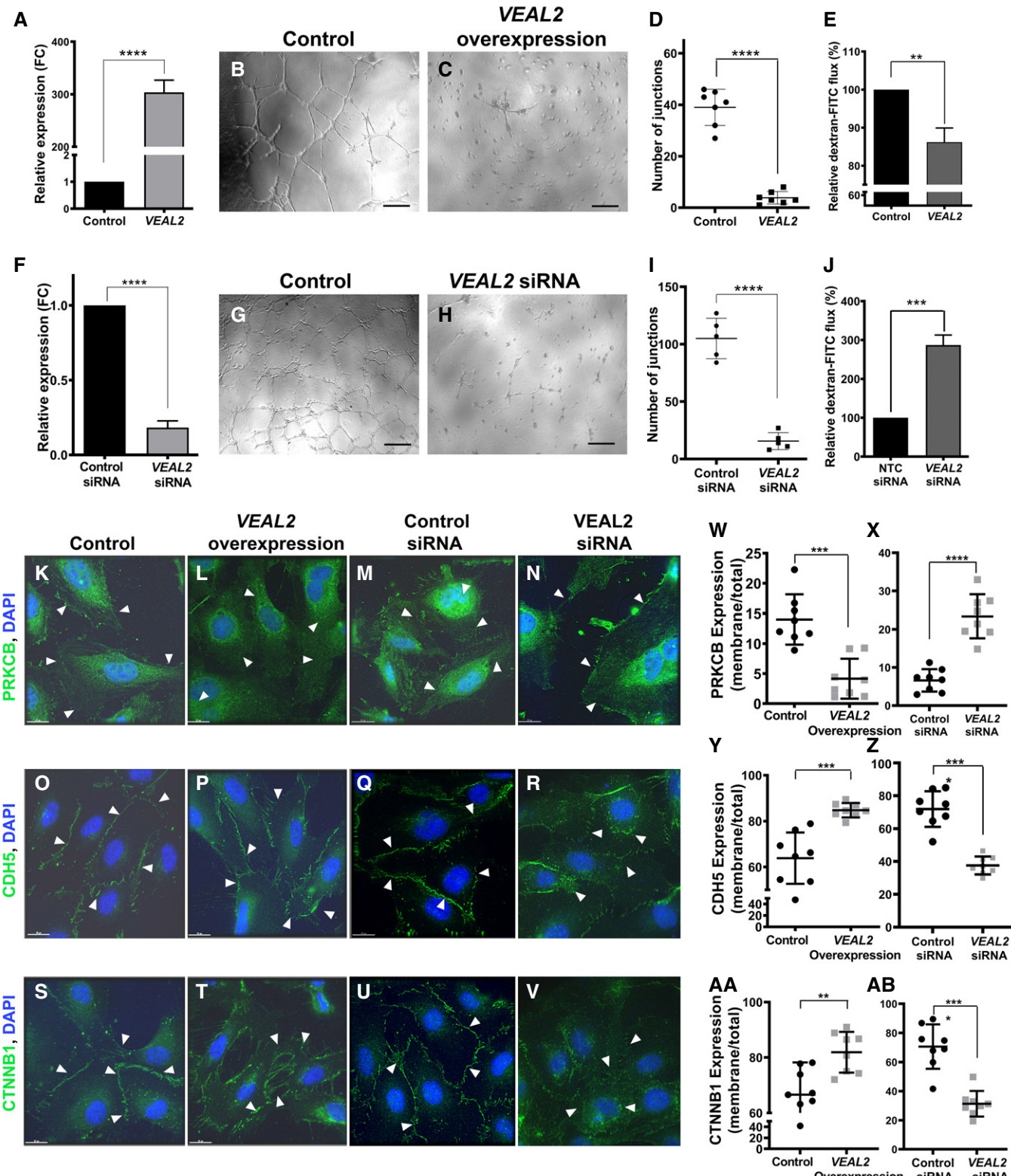

**Figure 6.**

clinically tested isozyme-specific PRKCB2 inhibitor, Enzastaurin (Bourhill *et al*, 2017), rescued hemorrhage significantly. Utilizing molecular simulation studies, we demonstrated that the C1 domain

binding lncRNA motif (CLM) in *veal2* interacts with residues H116 and S119 on the DAG-binding domain in Prkcbb, which was further substantiated by the competitive pattern of binding of human

PRKCB2 by *veal2* and DAG. The 8-base deletion in *veal2* putatively leads to alteration in RNA folding, disabling binding of CLM to Prkcbb presumably causing prolonged DAG-mediated Prkcbb activation. Using human PRKCB protein as bait in HUVECs, we identified an interacting uncharacterized lncRNA and human ortholog of *veal2*, named *VEAL2*. Rescue of *veal2*[gib005Δ8/+] embryos by complementing *VEAL2* indicated that human *VEAL2* and zebrafish *veal2* are functionally conserved. We demonstrated that *VEAL2* overexpression and knockdown in HUVECs alter endothelial junctional dynamics and permeability. Based on the finding of inhibition of PRKCB2 activity by *VEAL2*, we were keen to probe this mechanism in diseases associated with PRKCB2 activity. PRKCB2 is known to be hyperactive under diabetic conditions and causes various microvascular complications including retinopathy (Durpès *et al*, 2015). We analyzed *VEAL2* expression in patients with diabetic retinopathy (DR) and observed a reduced expression in the retinal choroid layer in DR patients compared to control individuals. *VEAL2* overexpression in the hyperglycemia model of HUVECs could recover excessive permeability phenotype by inhibiting translocation of PRKCB2 to cell membrane and disassembly of junctional complexes. Our results suggest that the orthologs *veal2* and *VEAL2* can efficiently regulate kinase activity of PRKCB2 and can act as potential biomolecules for targeting vascular diseases with PRKCB2 hyperactivation and elevated endothelial permeability.

Dissection of lncRNA mechanism *in vivo* has been typically attempted by deletion of whole lncRNA or promoter region, mostly in mouse and zebrafish (Sauvageau *et al*, 2013; Lavalou *et al*, 2019). However, large deletions of some lncRNAs have resulted in gross developmental defects (Sauvageau *et al*, 2013). Many GWAS and recent genomics studies have highlighted the relevance of small variations in lncRNAs in various diseases, including myocardial infarction, hereditary cataract, and thyroid carcinoma (Ohnishi *et al*, 2000; Pan *et al*, 2016; Eiberg *et al*, 2019). Small variations in lncRNAs impact structure-based functions as reported for MEG3 (Uroda *et al*, 2019). Interaction of Braveheart lncRNA with CCHC-type zinc finger nucleic acid-binding protein (CNBP) in cardiomyocytes was shown to be disrupted by an 11-base deletion within a G-rich RNA motif (Xue *et al*, 2016). We observe that a small 8-base

deletion in *veal2* can lead to endothelial dysfunction and hemorrhage phenotype. Disruption of secondary structure of *veal2* by a small deletion changes the folding and prevents the interaction between CLM of *veal2* and DAG-binding C1 domain of Prkcbb, causing endothelial integrity defects. We observed that haploinsufficiency of well-structured *veal2* in heterozygous *veal2*[gib005Δ8/+] zebrafish embryos causes hemorrhage due to vascular integrity defects. Haploinsufficiency of lncRNA causing specific phenotypes has been shown independently in a recent report where authors demonstrated that the haploinsufficiency of handsdown lncRNA in heterozygous mice exhibits hyperplasia in the right ventricular wall (Ritter *et al*, 2019).

The abundance in the transcriptome, coupled with spatio-temporally restricted and condition-specific expression patterns, makes lncRNAs strong contenders for biomarkers, drug targets, and biological substitutes of drugs themselves (Matsui & Corey, 2017). Novel functionally relevant lncRNAs have been discovered in model organisms and subsequently human equivalents have been identified using various approaches, even in the absence of sequence conservation (Johnsson *et al*, 2014). Recently, orthologs of ANRIL and JPX were identified on the basis of conserved structure and interacting partners, respectively (Cornelis *et al*, 2016; Karner *et al*, 2020). In our study, we demonstrate an approach where *VEAL2* orthologs in zebrafish and human were identified using their interacting evolutionary conserved protein partner PRKCB as bait, even in the absence of synteny or short conserved sequence stretches.

While studies have revealed the abilities of lncRNAs to liaise with proteins, RNA, and DNA through various mechanisms (Kopp & Mendell, 2018), a handful of lncRNAs have displayed potential to regulate kinases. NBR2 and CONCR are lncRNAs known to interact and regulate kinase activities of AMPK and DDX11, respectively (Liu *et al*, 2016; Marchese *et al*, 2016). NF-kappaB (NF-κB)-interacting lncRNA (NKILA) prevents overactivation of NF-κB, through a feedback loop (Liu *et al*, 2015). Our study shows that lncRNAs *veal2* and *VEAL2* prevent overactivation of PRKCB2 in endothelial cells by competitively binding to the DAG-binding domain. Although we could not differentiate two isotypes of PRKCB in the antibody based assays, RAP-MS data and *in vitro*

---

**Figure 7.** *VEAL2* is involved in diabetic retinopathy and can recover associated microvascular complications.

A Bar graph representing relative expression of *VEAL2* (in fold change) in choroid tissue isolated from control and diabetic retinopathy (DR) patients. Data obtained from 8 biological replicates (patients) and represented as individual values with mean fold change ± standard deviation.

B Bar graph representing relative fold change of *VEAL2* expression in blood samples of patients with different diabetic stages with aggravating vascular dysfunctions from diabetic mellitus (DM) to non-proliferative diabetic retinopathy (NPDR) to proliferative diabetic retinopathy (PDR) compared to control patients. Data were collected from 50 different patients in each condition and represented as individual values with mean fold change ± standard deviation.

C ROC curve shows sensitivity and specificity of *VEAL2* as a diagnostic biomarker for proliferative diabetic retinopathy with endothelial dysfunction.

D Complementation of *VEAL2* and *veal2* reverted increased permeability levels in the HUVEC monolayer model for hyperglycemia. Bar graph representing the effect of overexpression of *VEAL2* and *veal2* on permeability levels in hyperglycemia disease model, measured as efflux of dextran-conjugated FITC. Data obtained from 3 different biological replicates and plotted as mean percentage fold change values ± standard deviation.

E–S Modeling hyperglycemia in HUVEC resulted in dysregulation of junctional assembly of CDH5 and CTNNB1 proteins and increased membrane localization of PRKCB protein. Complementation of *VEAL2* and *veal2* in hyperglycemic conditions reverted junctional disassembly of CDH5 and CTNNB1 and also kept PRKCB in cytoplasm to mitigate pathological conditions associated with hyperglycemia. (E–H, Q) CDH5 protein, (I–L, R) CTNNB1 protein, and (M–P, S) PRKCB protein. (E–P) Magnification-60× and scale bar-15 μm. Arrowheads indicate representation of signals of proteins in HUVECs. (Q–S) Data from cells of different fields of 3 technical replicates of 1 biological replicate are presented as representation. Data are shown as individual values; the middle bar represents the mean, and the error bar represents ± standard deviation.

Data information: All the experiments $N \geq 3$. **$P$-value < 1E-2, ***$P$-value < 1E-3, and ****$P$-value < 1E-4. Statistics: (A) unpaired two-tailed $t$-test, (B, D, Q-S) one-way ANOVA with Bonferroni's multiple data comparison, and (C) Wilson/Brown method.

Source data are available online for this figure.

---

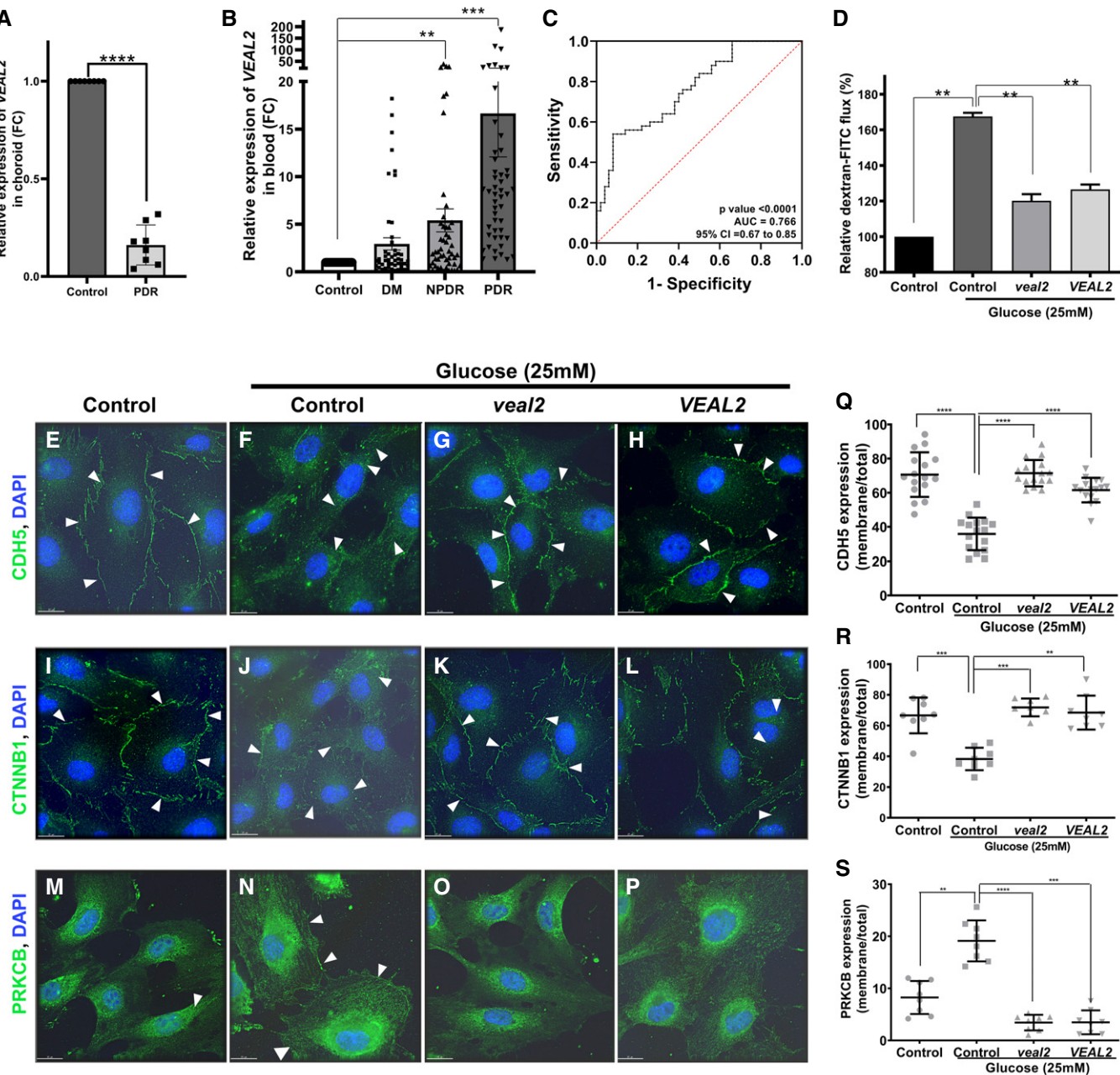

**Figure 7.**

kinase assays showed specific interaction of *VEAL2* with PRKCB2. Furthermore, *VEAL2* was detected at reduced levels in the choroid/vascular tissue and at elevated levels in blood taken from DR patients, where PRKCB2 is known to be hyperactive and is a tested drug target (Aiello *et al*, 1997). Elevated *VEAL2* levels in blood of DR patients could be due to leakage of *VEAL2* from unstable ECs of the vascular wall. From an earlier study, we found that *VEAL2* levels are elevated in exosomes released by endothelial cells (Pérez-Boza *et al*, 2018). So we speculate that hyperglycemia-induced exosomes from ECs could be a source of increased *VEAL2* levels in blood to communicate with other tissues/cells. Given the direct binding and near 50% inhibition of PRKCB2 by *VEAL2*, it

will be interesting to understand the potential of *VEAL2* in DR as a putative biomarker/drug target/adjuvant through future investigations. Although several PRKCB inhibitors have been tested in phase-III clinical trials, currently no approved drug is available in the market. PRKCB targeting poses the challenges of off-target effects and inefficacy in clinical settings owing to the presence of about 15 isoenzymes of protein kinase C family, necessitating for identification and characterization of isozyme-specific PRKCB inhibitors. The reported absence of tubulogenesis in Enzastaurin-treated ECs as tumor treatment parallelly suggests a potential application for *VEAL2* in inhibiting neo-angiogenesis (Spalding *et al*, 2008).

We induced hyperglycemia in HUVECs in order to model DR-associated endothelial pathophysiology, which was recovered by *VEAL2* overexpression. Ectopic expression of *VEAL2* prevented both hyperactivation of PRKCB2 and excessive junctional protein degradation. Further, the absence of any other interacting PRKC isoforms in *veal2* pulldown assay indicates isoform-specific inhibitory activity of *veal2* on Prkcbb. This implicates a potential therapeutic utility of *VEAL2* for modulating PRKCB activity in vascular diseases involving hyperpermeability. Other lncRNAs such as MIAT and MALAT1 have emerged as potential lncRNA candidates for drug targeting in DR (Michalik *et al*, 2014; Yan *et al*, 2015), while approaches to use lncRNA as biomarkers are being explored.

Based on earlier literature and novel data presented here, we propose a model where in basal conditions, *VEAL2* maintains levels of DAG-mediated activity of PRKCB in endothelial cells (Fig 8). Our results show that *VEAL2* in a competitive mode can compete with DAG to bind to the C1 domain of PRKCB2 and prevent its activation. A precise deletion in *VEAL2* in zebrafish or knockdown in HUVECs disrupts binding to PRKCB, causing extensive recruitment of PRKCB to the cell membrane. Overexpression of *VEAL2* in normal and hyperglycemic conditions retains PRKCB in cytoplasm, preventing translocation of junctional complexes from membrane to cytoplasm. However during knockdown of human *VEAL2* in HUVEC or upon a precise deletion in zebrafish *veal2*, the DAG-binding site in PRKCB is free for continuous binding and activation, leading to uncontrolled dissociation of endothelial junctional complexes. This further leads to a hyperpermeability condition in endothelial cells lacking *VEAL2*.

As a summary, in this study we discovered a novel lncRNA from zebrafish, identified its human ortholog, and further showed its implication in a human vascular disease. We observed excessive sprouting in zebrafish upon overexpression of *veal2* and also validated by increased *VEAL2* expression in newly formed vessels in fibrous layer in DR patients. Thus, *VEAL2* transpires to be an imperative regulator of angiogenesis in normal and pathogenic conditions. The levels of *VEAL2* are reduced in retinal choroid tissue of diabetic retinopathy patients compared to control individuals. Pathophysiology in both zebrafish lacking *veal2* and hyperglycemic HUVEC model was recovered by complementation of *VEAL2*, reverting the hyperactivation of PRKCB. This substantiates pivotal role of *VEAL2* in a well-deciphered and conserved endothelial pathway, and overexpression of *VEAL2* to inhibit PRKCB in endothelial cells (in a tissue/organ restricted manner) holds potential as a therapeutic strategy in managing vascular diseases involving hyperpermeability. Elevation in expression levels of *VEAL2* in blood of patients with diabetic retinopathy poses a potential application as a biomarker for aggravating microvascular complications. Taken together, this study provides evidence that *VEAL2* performs a hitherto unknown lncRNA-mediated regulation of PRKCB2 and junctional turnover for maintaining endothelial permeability.

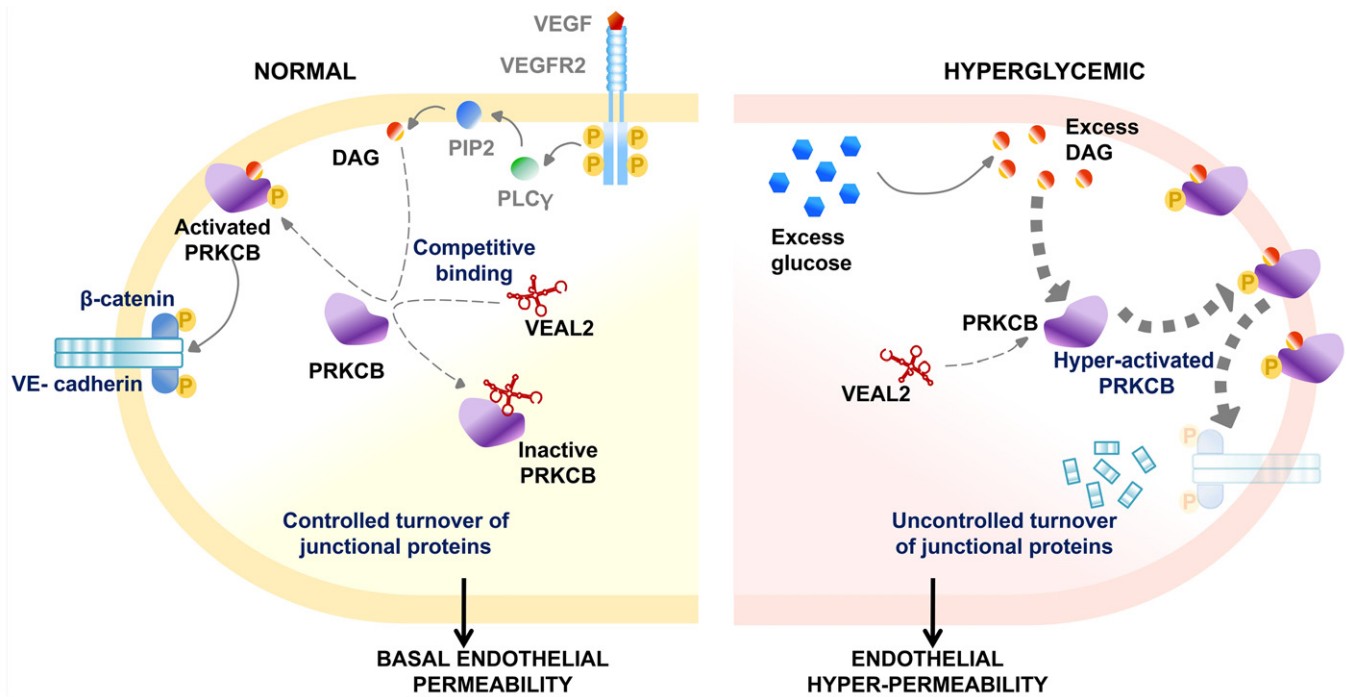

**Figure 8.  PRKCB-*VEAL2* interplay in maintenance of basal endothelial permeability.**

Hypothetical schema showing normal endothelial cell (left), in which the apparent competitive binding of DAG and *VEAL2* to the C1 domain of PRKCB restricts the activation of PRKCB and phosphorylation of junctional proteins. The controlled turnover of junctional proteins dictates the basal endothelial permeability and maintains homeostasis. Under hyperglycemic conditions (right), excessive glucose increases DAG levels in endothelial cells. Increased DAG levels further outcompete *VEAL2* and hyperactivate PRKCB, enabling increased translocation to membrane. During this condition, junctional protein turnover and degradation is unchecked and leads to hyperpermeability. VEGF-vascular endothelial growth factor, VEGFR2-vascular endothelial growth factor receptor 2, PLCγ-phospholipase C gamma, PIP2-phosphatidylinositol-4,5-bisphosphate, DAG-diacylglycerol, PRKCB-protein kinase C beta, and *VEAL2*-vascular endothelial-associated lncRNA 2.

# Materials and Methods

## Study approval

Double transgenic zebrafish (*Danio rerio*) *gib004Tg(fli1a:EGFP; gata1a:DsRed)* (Lalwani *et al*, 2012) and *veal2*$^{gib005\Delta8/+}$ were housed and handled according to the protocols and guidelines approved by the Institutional Animal Ethics Committee of CSIR-Institute of Genomics and Integrative Biology, India (GENCODE-C-24). Care was exercised to distress the animals minimally.

The study protocol for tissue and blood sample collection from patients with proliferative diabetic retinopathy or diabetes mellitus along with ethnically matched normal controls was approved by the Ethics Committee of the L V Prasad Eye Institute, India (LEC-02-14-029). The study protocol adhered to the tenets of the Declaration of Helsinki and was in accordance with good clinical/laboratory practices mentioned by the Ministry of Health and Family Welfare, Government of India. Written informed consent was obtained from subjects enrolled in this study.

## Statistics

Data are represented as mean ± SD of dependent samples. The comparisons were statistically tested by paired or unpaired t-test. For experiments with multiple group comparisons, analysis of variance followed by Bonferroni correction was performed. ROC curve was plotted using Δ Ct values with Prism-8.0. The *P*-values of < 0.05 were considered significant. Number of times experiment is performed is 3 or otherwise mentioned. All raw data and details on statistics applied are also given in supporting data for each figure.

## FACS-based isolation of endothelial and non-endothelial cells from zebrafish embryos

Double transgenic embryos of *gib004Tg(fli1a:EGFP;gata1a:DsRed)* (24–26 h post-fertilization (hpf)) were subjected to single-cell suspension preparation using trypsin–collagenase dissociation (Lalwani *et al*, 2012). The obtained single-cell suspension was used for fluorescence-activated cell sorting (FACS). Endothelial cells tagged with fli1-driven enhanced green fluorescent protein (EGFP) (GFP+) were sorted separately from unlabeled non-endothelial cells (GFP-). The endothelial cells formed roughly 5% of the total cell population from 24 to 26 hpf double transgenic embryos. The sorted endothelial cells (GFP+) and non-endothelial cells (GFP-) cells were pelleted down at 350 g and processed for RNA isolation.

## RNA isolation and sample preparation for RNA sequencing

The FACS sorted EC and NEC pellets were treated with TRIzol reagent (Ambion, USA). RNA was isolated subsequently using the RNeasy kit (Qiagen, USA) according to the manufacturer's protocol. Approximately 3 μg of RNA each from the sorted GFP+ endothelial and GFP- non-endothelial cells was used for sample preparation for RNA sequencing. Poly-A RNA capture was performed on the total RNA using Sera-Mag oligo (dt) magnetic beads (Thermo Scientific, USA). cDNA synthesis and second-strand synthesis were carried out using reverse transcriptase and DNA polymerase I (Invitrogen, USA), respectively, on the fragmented poly-A RNA. The 3' to 5'

exonuclease activity of the Klenow enzyme and the T4 DNA polymerase synthesis were carried out to repair the overhangs at the cDNA ends resulting in blunt end cDNA fragments. Klenow (3' to 5' Exon) was used to add a single A base overhang to the blunt end to facilitate base pairing with the manufacturer-supplied paired-end adapter (with a single T base overhang). Adapter-specific primers were used for the enrichment of the adapter-ligated products. The quality and quantity of the purified library was verified using agarose gel electrophoresis and Qubit (Life Technologies, USA).

## RNA sequencing and transcriptome assembly

Illumina cBOT cluster generation system was used to amplify the RNA libraries on the Genome Analyser IIx (GAIIx) flow cell. The amplified products were sequenced using sequencing by synthesis method on the Genome Analyser IIx (GAIIx), a sequencing platform from Illumina, USA. Base-calling was carried out for the obtained high-resolution images using Illumina pipeline software (v 1.9). Analysis was carried out on the reads that passed the quality filter. The sequencing reads obtained from EC and NEC were adapter-trimmed along with a quality cut-off of Q20 using Trimmomatic (Bolger *et al*, 2014). The reads were also sorted for length with a minimum length cut-off of 30 bases (SolexaQA) (Cox *et al*, 2010) and were aligned to the zebrafish reference genome (Zv9) using Tophat2 (Kim *et al*, 2013) which produced an alignment rate close to 82% in both samples. Cufflinks (Trapnell *et al*, 2012) were used to perform a *de novo* transcriptome assembly using default parameters. The transcriptomes from both samples were merged using Cuffmerge (Trapnell *et al*, 2012) and used for downstream analysis.

## Computational analysis pipeline for lncRNA identification

We defined transcripts longer than 200 nucleotides and lacking open reading frames longer than 100 amino acids as long non-coding RNAs (lncRNAs). The transcript sequences were downloaded from the UCSC Table browser. To begin with, transcripts longer than 200 bases were selected and "getorf" program from the EMBOSS (Rice *et al*, 2000) suite was used to predict ORFs within the transcript sequences. Tools such as Coding Potential Calculator (CPC) (Kong *et al*, 2007) and PhyloCSF (Lin *et al*, 2011) were employed to assess the coding potential of the transcripts with ORFs not longer than 100 amino acids. The non-coding transcripts thus extricated were overlapped with the RefSeq gene catalog to remove all protein-coding isoforms. In order to exclude transcripts with known protein domains, we employed HMMER3 (Eddy, 2009) to query the predicted ORFs against Pfam protein database (Finn *et al*, 2014) with default parameters and removed transcripts with significant Pfam hits (E-value < 0.001). The repeat-masked transcript sequences were subjected to BLASTx (Gish & States, 1993) against the zebrafish RefSeq protein database, and transcripts with an E-value < 0.0001 were excluded. BEDTools package (Quinlan, 2014) was used to classify the predicted lncRNA set into different classes based on their genomic context, by overlapping the lncRNA transcripts with well-annotated genes from RefSeq database. The lncRNAs were mapped to the promoter regions, defined as 5 kb upstream of a protein-coding TSS and classified them as promoter-associated lncRNAs. All the remaining lncRNAs were grouped as long intergenic lncRNAs (lincRNAs). The nearest protein-coding

gene to each lncRNA was also determined. BEDTools was further used to overlap the predicted lncRNA with the latest version of known lncRNA compiled from published database ZFLNC (Hu *et al*, 2018) (Accessed Oct 1, 2018), to generate the final novel endothelial and non-endothelial lncRNomes cataloged in this study.

## Ribosome profiling data analysis

Ribo-seq reads from 8 zebrafish developmental stages were downloaded from NCBI SRA (Chew *et al*, 2013). Adapter trimming was performed using fastx-clipper, part of FASTX-Toolkit (http://hannon lab.cshl.edu/fastx_toolkit/). Next, the reads mapping to rRNAs were removed after aligning the trimmed reads to zebrafish rRNA sequences downloaded from SILVA database using Bowtie2 (Langmead & Salzberg, 2012). In this step, 66% reads were discarded. About 235 million high-quality reads were isolated by placing a read length filter of 27–32 nt. These high-quality reads were then mapped using Tophat2 to the merged transcriptome assembly generated from this study and Zv9 reference genome, providing 95% alignment. The RNA-seq reads for the 8 stages of zebrafish embryos from NCBI SRA were downloaded separately. The reads were preprocessed using Trimmomatic and SolexaQA and mapped to the Zv9 reference genome using Tophat2. To compute Translation Efficiency Score (TES) for the novel lncRNAs, BEDTools was used to obtain the ribo-seq and RNA-seq read counts overlapping each lncRNA feature. The TES for each lncRNA was determined as the ratio of its ribo-seq read count to RNA-seq read count.

## qRT–PCR

RNA was isolated from the FACS sorted cells as described before. Superscript II (Invitrogen, USA) was used to synthesize cDNA from 1 μg of RNA (Peterson & Freeman, 2009). Quantitative real-time PCR was carried out using SYBR Green Mix (Roche, Germany) in LightCycler LC480 (Roche, Germany). *actb* was used as an internal control in all the experiments (if not mentioned separately) for the normalization. The primers used in the experiment are given in Appendix Table S3.

## 5' and 3' RACE

To determine the ends of the *veal2* lncRNA, we performed 5' and 3' RACE using 5' RACE System (Invitrogen, USA) and 3'RACE System (Invitrogen, USA) kits, respectively, and by following the manufacturer's protocol. Unique bands from the nested PCR of 5' and 3' RACE were gel-extracted and cloned in pCR 2.1-TOPO vector (Invitrogen, USA), and Sanger sequencing was performed on it using M13 primers. All primer details are given in Appendix Table S3.

## Ribosomal protein pulldown

ChIP grade antibody against ribosomal protein S12 (Rps12) (Abcam, UK) was used to pull down translated RNAs. Double transgenic *gib004Tg(fli1a:EGFP;gata1a:DsRed)* embryos at 24–26 hpf were subjected to single-cell suspension preparation as described in an earlier section. The cells were subjected to UV crosslinking using UV Stratalinker (Stratagene, USA) at optimal crosslinking settings. Cell lysate was prepared from crosslinked cells using RIPA buffer

(Sigma, USA). rps12b and IgG antibodies were incubated with 50 μl of protein A/G beads (Invitrogen, USA) in 1:50 proportion (of volume) for 2 h at RT. The protein A/G beads–antibody complex was further incubated with cell lysate overnight at 4°C with continuous mixing. Beads were washed thrice with a washing buffer (1XPBS + 0.1% Tween 20) and used to capture rps12b bound RNA species. cDNA was synthesized from the total RNA, and further selected lincRNAs were probed through end-point PCR.

## eGFP fusion constructs

In order to rule out the possibility of any functional peptide/ORF originating from *veal2* or *VEAL2*, eGFP fusion experiments were conducted. Full-length *veal2* and *VEAL2* sequences were amplified using Xho1-forward primer and kpn1-reverse primer. After digestion, amplicons were ligated into linearized eGFPN1 plasmid to clone *veal2* and *VEAL2* sequences in frame to N terminal of eGFP. IVT products were synthesized using T7 mMessage mMachine kit (Ambion, USA) and injected into single-cell zebrafish. Simultaneously, *mitfa-eGFP* (a kind gift from Dr. TN Vivek laboratory) was used as a positive control in the experiment. Primer details are given in Appendix Table S3.

## Whole-mount RNA in situ hybridization (WISH)

WISH was performed according to previously described protocol with minor modifications (Thisse & Thisse, 2008). Tris-buffered saline and Tween 20 (TBST) buffer was used as a washing buffer instead of PBST for stringent washing.

## Cell fractionation and RT–PCR assay

Cell fractionation was performed according to sucrose gradient centrifugation method as described earlier (Bhatt *et al*, 2012). Briefly, single-cell suspension was prepared from 24 to 26 hpf zebrafish embryos as mentioned in the above sections. The cell pellet was dissolved in 200 μl of cell lysis buffer (10 mM Tris–HCl (pH 7.5), 150 mM NaCl, 0.15% NP-40) and incubated on ice for 30 min with intermittent pipet mixing. Finally, cells were passed through a 27 G needle. Lysate was then layered on top of 2.5 volumes of chilled sucrose cushion (24% sucrose in cell lysis buffer) and centrifuged at 20,000 $g$ for 15 min at 4°C to separate the cytoplasmic fraction. The nuclei pellet was washed with 1× PBS-10 mM EDTA solution and resuspended in prechilled glycerol buffer (20 mM Tris–HCl (pH 7.5), 75 mM NaCl, 0.5 mM EDTA, 0.85 mM DTT, 0.125 mM PMSF, 50% glycerol). An equal volume of nuclear lysis buffer was added (10 mM Tris–HCl (pH 7.5), 1 mM DTT, 7.5 mM MgCl$_2$, 0.2 mM EDTA, 0.3 mM NaCl, 1 M Urea, 1% NP-40). The nucleoplasmic fraction was separated from chromatin fraction by centrifugation at 20,000 $g$ for 15 min at 4°C. One and nine parts of all the fractions were used for protein and RNA isolations, respectively. Proteins were precipitated using 10% (by volume) TCA and ran on 8% SDS–PAGE gel for quality check for all the three fractions. RNA was isolated from all three fractions using TRIzol (Invitrogen, USA) and RNeasy kit (Qiagen, USA) as recommended in standard protocols. The expression of *veal2* and *actb* (control) was checked using qRT–PCR as mentioned previously.

## Morpholino design and microinjections

An anti-sense morpholino oligonucleotide (MO) (Gene Tools, USA) (Appendix Table S2) was designed against the splice junction of *veal2* lincRNA. MO was dissolved in nuclease-free water (Ambion, USA) at a concentration of 1 mM according to the protocols recommended by Gene Tools, USA. The stock of 1 mM MO was stored at −20°C until further use. Working aliquots of MO oligos were prepared and stored at 4°C. About 3 nl of 500 μM MO was injected in 1-2 cell stage double transgenic zebrafish embryos. The animals were treated with 0.003% 1-phenyl 2-thiourea (PTU) in egg water. Animals with at least 50% post-injection survival rates were screened for vascular defects at 2 dpf. The effects of morpholino on the transcript were checked by intron spanning PCR. A pair of primers was designed on either exon, spanning the intron, and was used to amplify from cDNA from scrambled and *veal2* MO-injected animals.

## Complementation assays in zebrafish

For complementation assays in zebrafish mutant and *veal2* knockdown embryos, *veal2* wild-type RNA or its variants were synthesized *in vitro* (mMessage mMachine kit, Ambion) and injected into single-cell embryos collected from either control animals or the outcross of *veal2*$^{gib005\Delta8/+}$ animals. Injected animals were further screened for the phenotype to identify the level of rescue in complementation assays.

## Delivery of lncRNA into perivitelline cavity and alkaline phosphatase staining

Double transgenic *gib004Tg(fli1a:EGFP;gata1a:DsRed)* embryos were obtained and maintained in 0.003% PTU in egg water until 2.5 dpf, and all the embryos were manually dechorionated and anesthetized using 0.4% Tricaine prior to injection. Different dilutions of the *in vitro*-transcribed (IVT) *veal2* lncRNA product were prepared and injected at 3 nl volume into the perivitelline cavity of the anesthetized larvae. The injected larvae were then maintained in 0.003% PTU in egg water until 3.5 dpf, following which they were fixed using 4% paraformaldehyde (PFA) (in 1× PBS) and alkaline phosphatase assay was performed by the standard procedure (Nicoli & Presta, 2007).

## TALEN design, microinjections, and screening

To achieve the deletion of the entire lncRNA, two pairs of TALENs were designed around the locus of the *veal2* lncRNA. The TALEN target sites were predicted using the open free access software mojo hand (Neff *et al*, 2013). The candidates which gave a single hit across the genome were used further for building the TALENs using a modified version of the Golden gate cloning strategy (Cermak *et al*, 2011). An equimolar cocktail of 25 pg per arm of the TALENs was injected into single-cell zebrafish embryos by standard protocols. Animals were treated with PTU (0.003% in egg water) to inhibit melanin pigment formation. Phenotypic analyses were performed only if at least 50% post-injection survival rates were at 2 dpf.

DNA was isolated from single phenotypic embryos as described earlier (Davidson *et al*, 2003). Primer pairs (Appendix Table S3) were designed flanking TALEN target sites, and PCR was performed. Gene deletions were detected through Sanger sequencing of eluted products (Bioserve Biotechnologies, Hyderabad, India).

Single TALEN pair dose standardizations were performed by injections into one-cell staged transgenic embryos. An optimal dose of 25 pg per TALEN arm was injected into transgenic embryos by standard protocols. Animals were treated with PTU (0.003% in egg water), and phenotypic analyses were performed at 2 dpf.

## O-dianisidine staining

For the identification of localization of hemoglobin containing cells in zebrafish, embryos were stained with O-dianisidine stain as described previously (Lieschke *et al*, 2001). MO-injected animals or TALEN-injected animals and control animals were incubated in O-dianisidine stain solution (0.6 mg/ml O-dianisidine (Sigma-Aldrich, USA), 40% ethanol (Merck, USA) with 0.01 M sodium acetate, 0.65% $H_2O_2$) for 15 min in dark. Embryos were washed with 1× PBS thrice before imaging.

## Heteroduplex mobility shift assay (HMA)

In order to identify indels at the single TALEN target site, primers (Appendix Table S3) were designed around the target site to generate an amplicon of around 200 bp. The region was PCR-amplified, and the amplicons were subjected to denaturing and slow cooling as described previously, to generate heteroduplexes (Ota *et al*, 2013). The amplicons were further run on a 15% non-denaturing DNA PAGE gel. The heteroduplexes were excised, eluted by cut and soak method (Sambrook & Russell, 2006), and subjected to Sanger sequencing.

## Generation of *veal2*$^{gib005\Delta8/+}$ line

Single TALEN-injected phenotypic animals (F$_0$) were raised to adulthood. In order to identify their genotypes, fin clipping was performed, followed by DNA isolation as mentioned in an earlier section. The targeted locus was amplified and probed using HMA, followed by Sanger sequencing. Positive animals harboring mutations were outcrossed to determine the transmission of the editing event to the next generation (F$_1$). Phenotypic animals were subjected to genotypic screening, and the rest of surviving phenotypic animals were grown to adulthood. Adult F$_1$ animals were fin-clipped and subjected to genotype screening using HMA.

In order to identify potential off-targets in F$_2$ animals, PROGNOS tool (Fine *et al*, 2014) was used to predict off-target sites with "Max Mismatches per half-site" set to 4. Primers (Appendix Table S3) were designed around the predicted off-targets and subjected to HMA, followed by Sanger sequencing. A single F$_1$ line with an 8 bp deletion at positions 28 to 33 bases of the *veal2* transcript (*veal2Δ8*) was further propagated to generate a stable *veal2* knockout line.

## Electro-mobility shift assay (EMSA)

To detect the protein binding affinity of *veal2*, electrophoretic mobility shift assay (EMSA) was used. Protein lysate from 24 to 26 hpf zebrafish embryos was prepared using NP-40 lysis buffer (Sigma, USA). 1–5 mg of total protein lysate and 1 μg of digoxigenin-labeled

*veal2* IVT were incubated at 28°C for 1 h before probing. The standard protocol mentioned by DIG Gel Shift Kit, 2$^{nd}$ Generation, Roche, was used to detect the mobility shift.

## RNA anti-sense pulldown, LC-MS, and Western blotting

For anti-sense pulldown, PAGE-purified 5' biotinylated 50 mer probes (Appendix Table S3) (Integrated DNA Technologies, USA) were used. Double transgenic *gib004Tg(fli1a:EGFP;gata1a:DsRed)* embryos at 24–26 hpf were subjected to single-cell suspension preparation and UV crosslinking as described in an earlier section. Crosslinked cells were further used for RAP-MS as described previously (McHugh *et al*, 2015). For LC-MS, samples were detected with Acquity Waters UPLC system (Sandor Life Sciences, Hyderabad, India) using following criteria: 2.1 μm × 100 mm × 1.7 μm CSHC18 column was used, and samples were separated for 60 mins using buffer A: 0.1% formic acid and buffer B: acetonitrile with 0.1% formic acid. For detection of the peptides, data were analyzed using PLGS search engine (Waters, USA). The following parameters were set to predict peptides: the peptide tol (ppm): 30; fragment tol (ppm): 70; modification: carbamidomethyl_c,oxidation_m; missed cleavage: 1; and database used for analysis: Danio rerio (UniProt). After the prediction of peptides from different samples, the common peptides across all three biological replicates with at least 1.5-fold change across *veal2* pulldown and control (no probe) pulldown samples were identified. Further, proteins involved in the angiogenesis pathway were predicted using PANTHER Gene Ontology (GO) analysis tool (Mi *et al*, 2019) and prioritized for validation using Western blotting. Anti-PRKCB (ab189782), anti-DeltaC antibody (ab73336, Abcam), anti-DeltaD antibody (ab73331, Abcam), and anti-EPHB3 (ab171519) antibodies were used for Western blotting. Anti-PRKCB antibody is not isozyme-specific to differentiate Prkcbb or Prkcba. So to be accurate in terminology, we have used PRKCB instead of PRKCB2 in experiments involving Anti-PRKCB (ab189782) antibody.

## RNA immunoprecipitation (RIP)

For RNA immunoprecipitation (RIP), double transgenic *gib004Tg (fli1a:EGFP;gata1a:DsRed)* embryos at 24–26 hpf were subjected to single-cell suspension preparation and UV crosslinking as described in an earlier section. Crosslinked cells were further used for RIP as described previously with some modifications (Li *et al*, 2014). Briefly, 60 μl/ sample of protein A magnetic beads (Invitrogen, USA) were taken and washed thrice with PBST. Washed beads were allowed to incubate with 1.2 μl of primary antibody (1:50 by volume) in 60 μl of PBST solution for 2 h at room temperature. Beads were further pulled by placing on a magnetic stand and were washed thrice using PBST. After washing, beads were resuspended in the cell lysate, followed by overnight incubation at 4°C. After incubation, beads were pulled and washed thrice using 1XPBST solution and divided in 1:4 parts for isolation of protein and RNA, respectively. RNA was isolated using the standard TRIzol method and further used for qRT–PCR.

## *In vitro* PRKCB2 kinase assay

For *in vitro* kinase assay, ADP-Glo kit (Promega, USA) and purified PRKCB2 protein (Promega, USA) were used. The standard protocol

given by the manufacturer was used for the assay. About 52 mM of the PRKCB2 protein was used per reaction. To test the competitive inhibition of PRKCB2 by *veal2* and DAG, different dosages of each were used. Lipid activator (0.5 mg/ml phosphatidylserine, 50 μg/m 1-stearoyl-2-linoleoyl-sn-glycerol, 50 μg/ml 1-oleoyl-2-acetyl-sn-glycerol, 0.15% Triton X-100, 1 mM DTT, 2 mM CaCl$_2$, 20 mM MOPS, pH 7.2) (SignalChem, USA) (10X–100 μM) was concentrated ten times using vacuum concentrator (Eppendorf, USA) and used to prepare dilutions. Different dosages of lipid activator (DAG) (No DAG, 1, 2, 4, 6, 8, 10, 50, 100 μM) with or without different concentrations of *veal2* (0.025, 0.25, 2.5 nM, 25 mM) were tested in PRKCB2 kinase assay.

## Establishing the structure of the protein kinase C beta

To elucidate the high-resolution structure of the zebrafish protein kinase C beta (Q7SY24) and human protein kinase C beta (P05771) proteins, the initial model was first predicted using Phyre2 (Kelley *et al*, 2015). Subsequently, molecular dynamic simulation was performed using GROMACS 4.6.1 (Van Der Spoel *et al*, 2005). All atomistic simulations were carried out using the CHARMM36 all-atom force field (Vanommeslaeghe *et al*, 2010; Huang *et al*, 2017) using periodic boundary conditions. The starting model was solvated in a periodic box with 97916 TIP3 water molecules. A 26 Na ions were added to the solvent to neutralize the electrical net charge of the protein. The system was then minimized for 50,000 steps using a steepest descent algorithm with an emtol value of 200 KJ/mol after a minimization with emtol of 100 KJ/mol. This was followed by an equilibration run of 100 ps in the NVT ensemble with restraints on the protein atoms. The NPT ensemble was used for production simulation. Systems were simulated at 300 K, maintained by a Berendsen thermostat with a time constant of 1 ps with the protein and non-protein molecules coupled separately. Pressure coupling was done employing a Parrinello–Rahman barostat using a 1 bar reference pressure and a time constant of 2.0 ps with compressibility of 4.5e-5 bar using isotropic scaling scheme. Electrostatic interactions were calculated using the Particle Mesh Ewald (PME) summation. The production run was performed for 1 μs.

## Establishing the structure of *veal2* and *VEAL2*

For 3D structural analysis, the optimal secondary structure in dot-bracket notation was obtained from RNAfold WebServer (Hofacker, 2003), Gruber *et al*, 2008) by calculating the partition function and base pairing probability matrix in addition to the minimum free energy (MFE) structure. MFE was calculated using a loop-based energy model (using Turner model for Energy Parameters) and McCaskill's algorithm (Zuker & Stiegler, 1981) for the secondary structures contributing toward the minimum free energy in the RNA by summing the contributing free energies from the loops.

The dot-bracket notation was then used as structural constraints in the MC-Fold-MC-Sym pipeline (Parisien & Major, 2008). Structural constraints force certain nucleotides to be either paired or unpaired and will restrict the conformational search space. The advantage of using a dual approach is that it shall use the best secondary model from the first method and feed it as a template to guide and further predict the new secondary structure using the MC-Fold algorithm. The final tertiary structure of *veal2* was predicted

using the MC-Fold MC-Sym pipeline. The energy-minimized model was obtained using the method's scoring function, which calculates the base pairing energy contribution by reducing the nucleotides into cyclic motifs.

### Interaction between Prkcbb and veal2 or VEAL2

Structural details of the interaction between Prkcbb and veal2, veal2Δ8, or VEAL2 were established by docking using HDOCK (Huang & Zou, 2014; Yan et al, 2017) a hybrid algorithm that uses template-based as well as ab initio docking.

### Cell fractionation for cytosolic and membrane fractions

For separating 2 different fractions of the cells: cytoplasmic and membrane, high-speed centrifugation based method was used. Single-cell suspension of 2 dpf control and $veal2^{gib005Δ8/+}$ embryos was prepared as described in an earlier section. Cell pellet was dissolved in RIPA buffer (Sigma, USA) for total cell lysate and fractionation buffer (20 mM HEPES pH 7.4, 10 mM KCl, 2 mM $MgCl_2$, 1 mM EDTA, 1 mM EGTA, 1 mM DTT, 1X PIC) for isolating different fractions. After incubation on ice for 20 min, cells were passed through a 27 G needle and centrifuged at 700 g for 5 min at 4°C. The supernatant was further centrifuged at 10,000 g for 5 min. The pellet contained mitochondria and other dense organelles. The supernatant was used for high-speed centrifugation at 100,000 g for 45 min 4°C. The pellet containing a membrane fraction was washed using a fractionation buffer. The pellet was dissolved in 1XTBS buffer containing 0.1% SDS, and the cytoplasmic fraction was concentrated to 50–75 µl. For total cell fraction, after cell lysis using RIPA buffer, the lysate was centrifuged at 10,000 g for 15 min 4°C. Externally 1.5 ng of purified GFP protein was spiked in all fractions for normalization.

### Enzastaurin treatment and rescue of veal2 mutants

Initially, a dosage titration of Enzastaurin (Sigma, USA) across 1, 5, 10, and 50 µM was used for treatment of zebrafish embryos from 8 hpf to 36 hpf time window. After the dose standardization, 10 µM was used to treat veal2 mutant embryos from 8 hpf to 36 hpf. After 48 hpf, embryos were processed for O-dianisidine staining as mentioned previously. O-Dianisidine-stained embryos were imaged and then scored for the hemorrhage phenotype.

### RIP-seq of PRKCB2 in HUVECs

To understand the RNA molecules interacting with PRKCB2 in HUVECs, 2 M cells were UV crosslinked and processed as mentioned above in the RIP section. After the RNA capture, RNA libraries were prepared using a stranded TruSeq RNA library kit (Illumina, USA) as per the manufacturer's protocol. Prepared libraries were sequenced using Hiseq-2500 (Illumina, USA) and analyzed for transcript prediction as mentioned in the above section.

### Construction and transfection of lncRNA VEAL2 and veal2 overexpression plasmid

The plasmid pcDNA 3.1+ was used to clone full-length VEAL2 or veal2 using EcoRI and NotI restriction sites. The EcoRI and NotI restriction enzyme sites were introduced at the ends of full-length VEAL2 or veal2 in vitro using specific primers (primer details given in Appendix Table S3). Then, this fragment was cloned into a linearized pcDNA 3.1+ vector. Finally, the sequence of the identified recombinant plasmid was confirmed by Sanger sequencing. The transfection was performed using HUVEC nucleofector kit (vpb-1002) (Lonza, USA) according to the manufacturer's protocol as follows: (i) HUVECs were cultured in EGM2 media (Lonza, USA) (P2-P5) in T-75 flasks at 37°C and with 5% $CO_2$. (ii) 1 M cells were pelleted down and dissolved in 100 µl electrolyte in a cuvette and nucleofected using A-034 program in nucleofector machine from Lonza, USA. (iii) After nucleofection, cells were dissolved in additional 900 µl HUVEC EGM-2 media and aspirated to a fresh tube. These transfected cells were further used for various assays.

### Transfection of siRNA targeting VEAL2 in HUVECs

In order to design siRNAs, we used Dharmacon's siDESIGN and ordered 4 siRNAs (Dharmacon, USA). Initially, we transfected all 4 siRNAs independently and in cocktail at final concentration of 100 nM in HUVECs using HUVEC nucleofector kit (vpb-1002) (Lonza, USA) as per mentioned above. siRNA3 was most efficient, and we further used it for various assays. Sequence is given in Appendix Table S2.

### Migration, angiogenesis, and permeability assays of HUVECs

In all assays measuring migration, angiogenesis, and permeability, cells were nucleofected by control plasmid or lncRNA-VEAL2 or veal2 overexpression plasmids or control siRNA or VEAL2 siRNA using methods as mentioned earlier (Abdullaev et al, 2008; Shinde et al, 2013). For analysis of migration, scratch assay was performed. The wound was imaged and measured at 0, 3, 6, 9, and 24 h post-scratch. The wound closure rate was detected using analysis of different images from different time point using ImageJ software. For angiogenesis detection, Matrigel (BD Biosciences, USA) was thawed overnight at 4°C and 75 µl of Matrigel and 50 µl of EGM2 media (Lonza, Switzerland) was added per well of 48-well plate and incubated at 37°C for 30 min. The above treated cells were re-seeded on Matrigel-coated wells at a density of 50,000 cells per well. After 24 h of incubation at 37°C, 4-5 field of views (magnification 5×) were selected per well to analyze the tube formation and number of junctions between vessels using Angiogenesis Analyzer tool in ImageJ (Carpentier et al, 2020). To detect the permeability, after treatment, cells were grown on Matrigel-coated 6-well filters (Corning BioCoat Matrigel, Corning, USA) at a density of 0.1 M cells per well. Cells were grown for 3 days until confluency forming a tight monolayer. Permeability of the formed endothelial monolayer was detected using dextran conjugated with fluorescein isothiocyanate (FITC) (Sigma, USA). Dextran-FITC at 1 mg/ml final concentration was added on top of the filter, and 50 µl media in duplicates were collected from lower well at 0, 15, 30, 45, 60, 90, and 120 min. The number of molecules of dextran-FITC passed through endothelial monolayer was estimated by measuring fluorescence of media at 480 nm.

## Immunofluorescence in HUVECs

To detect the localization of CDH5, CTNNB1, and PRKCB, immunofluorescence in HUVECs was conducted as previously shown (Shinde *et al*, 2013). In short, treated cells were cultured on coverslips in 6-well plates for 24 h and fixed using 4% paraformaldehyde (Sigma, USA) treatment for 15 mins. The cells were washed thrice with 1XPBS and followed by treatment with 1XPBS+0.3%Triton X for 15 mins for permeabilization. After blocking (1XPBS+ 0.1% Triton X + 5%BSA), cells were incubated with primary antibody (PRKCB-ab189782, CDH5-ab33168, CTNNB1-ab6302, Abcam, USA) diluted 1:50 in 1XPBS + 0.1% Triton X + 2% BSA for 2 h at room temperature. The cells were washed thrice with 1XPBS+ 0.1% Tween 20. Further, the cells were incubated with anti-rabbit secondary antibody conjugated with Alexa Fluor 488 at 1:500 dilution in 1XPBS + 0.1% Triton X + 2% PBS and incubated for 2 h. After incubation, cells were washed thrice with 1XPBS + 0.1% Tween 20 and mounted on slides with DAPI+mounting media (F6057, Sigma) and visualized under 60× using DeltaVision microscope (GE healthcare, USA).

## Single molecule fluorescence *in situ* hybridization (smFISH)

To detect the subcompartment localization of *VEAL2* in HUVECs, smFISH was performed as per the manufacturer's recommendations for Stellaris FISH protocol for adherent cells (Biosearch Technologies). Some modifications were adapted in our study. We used double concentration of the probe (final conc. 250 nM) in hybridization buffer, and cells were incubated for 16 hrs for hybridization. After washing thrice, coverslips were mounted on slides with DAPI+mounting media (F6057, Sigma) and visualized under 60× using DeltaVision microscope (GE healthcare, USA).

## Co immunofluorescence and smFISH

To detect the *in vivo* interaction of *VEAL2* and PRKCB, we performed Co-IF and smFISH as per recommendation for Sequential Stellaris FISH and Immunofluorescence using Adherent Cells by manufacturer (Biosearch Technologies).

## Human DR patient details enrolled in the study

Expression analysis of *VEAL2* was performed in different tissues isolated from diabetic retinopathy (DR) patients and control samples. Choroid tissues were obtained from the eyes of DM patients with early DR changes (age: 75.33 ± 5.04) with unknown diabetic duration and controls (age: 76 ± 4.61 years). Blood samples were collected from proliferative diabetic retinopathy (PDR) patients (age: 53.86 ± 1.61 years) suffering from diabetes for 15.05 ± 0.90 years and controls (age: 65.84 ± 1.03 years). The epiretinal membranes containing fibrous layers were isolated from the PDR patients (mean age: 60.5 ± 9.394 years) and controls (mean age: 56 ± 5.066 years). The controls were patients who did not have a history of diabetes and were operated for macular hole or retinal detachment.

## Expression analysis of *VEAL2* in DR patients

RNA was isolated from the collected blood samples using TRIzol method and retinal tissues and epiretinal membranes using Pure-Link kit (12183018-A).

## Hematoxylin staining of retina

Cadaveric control and diabetic eyes were collected in a sterile moist chamber within 24 h of death from the Ramayamma International Eye Bank at the LV Prasad Eye Institute, Hyderabad, India, according to the tenets of Declaration of Helsinki. The retinal tissues were carefully removed from the eyes under a dissection microscope and kept in 4% formalin for fixation and paraffin sections were made. Retinal tissue sections were deparaffinized at 60°C for 20–30 min followed by two washes of xylene (Catalog No. 40575, Sd Fine-CHEM limited) for 10 min. Sections were dehydrated by immersing twice in absolute alcohol for 5 min. Subsequent immersion was done in 95 and 70% alcohol for 2 min each. After a brief wash in distilled water, sections were stained in Harris hematoxylin (Catalog No. H3136, Sigma) solution for 15–20 min followed by washing under running tap water for 5 min. Sections were differentiated in 1% acid alcohol for 30 s. The sections were washed under running tap water for 5 min followed by rinsing them in 95% alcohol and counterstained with eosin–phloxine solution for 30 s to 1 min. Finally, the sections were dehydrated by dipping them in 95% alcohol twice for 5 min. Finally, these sections were dipped twice in xylene for 5 min and mounted using mounting DPX (Catalog No. POICHA-R-391780, Sd Fine-CHEM limited) medium. Images were taken at 20× using OLYMPUS-BX51 microscope and analyzed for pathological changes in the diabetic and control tissues.

## Imaging

The control and injected embryos were observed and imaged with an upright Zeiss Axioscope A1 fluorescent microscope (Carl Zeiss, Germany) and SP8 TCS confocal microscope (Leica, Germany) at 2 dpf. Immunofluorescence in HUVEC monolayer was imaged using DeltaVision microscope (GE Healthcare). The images were processed with Zeiss AxioVision 4.6 or Leica LAS X and DeltaVision Ultra from GE healthcare.

## Nomenclature of *veal2*/*VEAL2*

We have followed the nomenclature guidelines provided by a recent article published in EMBO (https://www.embopress.org/doi/full/10.15252/embj.2019103777). Zebrafish vascular endothelial-associated lncRNA is termed as "*veal2*", and its functional ortholog in human endothelial cells is named as "*VEAL2*".

# Data availability

The raw data for RNA sequencing of EC and NEC populations in zebrafish and RIP-seq data of PRKCB in HUVECs have been deposited in the NCBI Sequence Read Archive (SRA) under the accession PRJNA504385 (https://www.ncbi.nlm.nih.gov/bioproject/PRJNA504385/).

**Expanded View** for this article is available online.

## Acknowledgments

The authors thank Drs. Souvik Maiti, Vivek Natarajan, and Manikandan Subramanian for critical comments on the manuscript. We acknowledge the contributions of Ankit Verma and Rijith Jayarajan for help with RNA sequencing experiments. We thank Drs. Kriti Kaushik and Shruti Kapoor for help with the initial bioinformatic analyses. We thank Shadabul Haque for his contributions toward the morpholino standardization experiments in zebrafish and Yusman Manchanda for help with maintaining zebrafish lines. The authors acknowledge kind gifts from Dr. Kausik Chakraborty for GFP protein and Dr. Vivek Natarajan for *mitfa-e-GFP*-cloned plasmid. The authors also acknowledge Mr. Manish Kumar for technical assistance in confocal microscopy. This work was generously supported by the Council of Scientific and Industrial Research (CSIR), India, through the grants BSC0123, MLP1801, and BSC0403 for imaging facility. PS, SM, and AS acknowledge their Senior Research Fellowships from CSIR, India and SV acknowledges support from Manipal Academy of Higher Education in Manipal, India towards her PhD. RKM acknowledges Regional Centre for Biotechnology for Institutional Core funding and Science & Engineering Research Board (SERB) Start-up Research Grant (SRG/2019/000495), Department of Science and Technology, India. IK and SC were supported by the Department of Biotechnology, India (BT/01/COE/06/02/10 and BT/PR32404/MED/30/2136/2019).

## Author contributions

PS, VS, and SSB conceived and designed the study. PS, SM, JT, SV, GR, KVS, RCB, AP, EL, ML, and AV performed the NGS, molecular biological, and biochemical experiments. PS, SM, RRP, MT, SJ, BBK, SC, IK, RKM, and SSB analyzed the data and interpreted the results. AS and VS performed the bioinformatic analyses. AR and SSG performed molecular simulation studies. PS, SM, AS, VS, and SSB wrote the manuscript.

## Conflict of interest

The authors declare that they have no conflict of interest.

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
