## [Review Process File · The EMBO Journal]

LncRNA VEAL2 regulates PRKCB2 to modulate endothelial permeability in diabetic retinopathy

Paras Sehgal, Samatha Mathew, Ambily Sivadas, Arjun Ray, Jyoti Tanwar, Sushma Vishwakarma, Gyan Ranjan, KV Shamsudheen, Rahul Bhojar, Abhishek Pateria, Elvin Leonard, Mukesh Lalwani, archana vats, Rajeev Pappuru, Mudit Tyagi, Saumya Jakati, Shantanu Sengupta, Binukumar BK, Subhabrata Chakrabarti, Inderjeet Kaur, Rajender Motiani, Vinod Scaria, and Sridhar Sivasubbu **DOI: 10.15252/emboj.2020107134**

Corresponding author(s): Sridhar Sivasubbu (sridhar@igib.in) , Vinod Scaria (vinods@igib.in)

Review Timeline:

Submission Date:	6th Nov 20
Editorial Decision:	27th Nov 20
Revision Received:	4th Apr 21
Editorial Decision:	4th May 21
Revision Received:	16th May 21
Accepted:	21st May 21

Editor: Ieva Gailite

Transaction Report:

Thank you for submitting your manuscript for consideration by the EMBO Journal. We have now received three referee reports on your manuscript, which are included below for your information.

As you will see from the comments, all reviewers find the presented identification of two lncRNAs targeting PRKCB in zebrafish and human endothelial cells interesting. However, they also indicate a number of important concerns that affect core conclusions of the study, which would have to be addressed before they can support publication of the manuscript. In particular, they ask to clarify H-VEAL2 expression levels in human endothelial cells, to provide further evidence to the proposed regulation of PRKCB activity and membrane localisation by zebrafish and human lncRNAs, and to clarify potential functional differences between the two lncRNAs.

Based on the interest expressed by the reviewers, I would like to invite you to address the comments of all referees in a revised version of your manuscript. Please note that a strong referee support will be required for the acceptance of the revised manuscript.

I should add that it is The EMBO Journal policy to allow only a single major round of revision and that it is therefore important to resolve the main concerns at this stage. I would be happy to discuss the revision in more detail via email or phone/videoconferencing. Please also let me know if you find that particular issues will not be addressable in the revised version, in which case I would be happy to discuss alternative publication possibilities.

We have extended our 'scooping protection policy' beyond the usual 3 month revision timeline to cover the period required for a full revision to address the essential experimental issues. This means that competing manuscripts published during revision period will not negatively impact on our assessment of the conceptual advance presented by your study. Please contact me if you see a paper with related content published elsewhere to discuss the appropriate course of action.

Please feel free to contact me if you have any further questions regarding the revision. Thank you for the opportunity to consider your work for publication. I look forward to receiving your revised manuscript.

Referee #1:

The authors describe a pair of previously unstudied lncRNAs, in zebrafish and in human, that play important roles in the biology of endothelial cells. They also identify and report the mechanism of action of these lncRNAs, that bind and modulate the activity of the PRKCB protein involved in a signalling cascade important in the endothel. Overall, the study is very well done and should be of high interest to both the lncRNA community and the endothelial cell community, and so fits well the scope of the journal. The data on zebrafish is overall solid and convincing. I do have some concerns about the human data, as detailed below.

Major comment:

- It is not clear from the existing and presented data how robustly/highly expressed is H-VEAL2. For the proposed activity it appears that it should be expressed at rather high levels. However, in ENCODE HUVEC RNA-seq data, there are barely any reads on the strand from which H-VEAL2 is

transcribed, and the FANTOM CAGE evidence for a promoter there is also very weak. In GTEx data, it seems to be primarily expressed in the skin. Can the authors show RNA-seq tracks (from their RIP, and ideally from some strand-specific protocol) that show that the transcript is indeed originating from the antisense region of MYADM, and indeed spliced as in the gene model? How abundant is it in HUVEC cells? These points are crucial to make the proposed functionality of H-VEAL2 convincing.

Minor comments:

1. Line 63, typo: "specialized epithelial cells that forms"
2. Lines 207-213, the part about Enzastaurin is confusing. The treatment by Enzastaurin does not cause any phenotype by itself? So the idea is that blocking the ATP-binding catalytic domain prevents VEAL-2 from accessing it? This needs to be more clearly explained. There is also a typo in "Enz*u*staurin" in Fig. 3E.
3. Line 223: Does mutation of motif-3 affect the ability of VEAL-2 to inhibit PRKCB in vitro?
4. Line 281: Please include a KD control for the FISH of H-VEAL2 (in light of the concerns above)

Referee #2:

The present study reports a novel endothelial expressed lncRNA, named *veal2*, which control vascular permeability in zebrafish. The authors further report that *veal2* interact with *Prkcbb* to limit its enzymatic activity. Pharmacological inhibition of *Prkcbb* prevented *veal2* mutant phenotypes in zebrafish. The strength of the study is that authors report a phenotype of the *veal2* mutant which is rescued by zebrafish and human *Veal2*. However, the study has several weakness as outline below.

1. It is unclear why the 8 bp deletion was chosen as *zveal2* loss of function model. In the follow up studies, the authors identified motif 3 as putative interaction motif. If this is true, motif 3 mutations should be studied in zebrafish. In addition, the biochemistry supporting Figure 4 is weak and the relevance of the interaction motifs should be further supported by in vitro experiments showing the effects of *z-veal2* deletion/point mutations on enzyme activity. Adding human *Veal2* / mutants would also help to confirm that the human lncRNA, which according to the authors lacks sequence and structural homology, is indeed a functional orthologue.
2. The effects of *veal2* on zebrafish and human EC function are not fully compatible since endothelial cells in culture show a profound anti-angiogenic response (likely also toxicity), which is not observed to that extent in zebrafish. This might be in part related to the lack of conservation or vascular bed specific actions of *Veal2*. It may help to provide a more detailed analysis of vessel morphology (including high magnification images) in the zebrafish.
3. In addition, the in vitro studies are not very convincing and have several technical and conceptual weaknesses: i) the control plasmid and control siRNA transfected cells show major differences in baseline values. Unspecific effects of the transfection protocols should be taken into account and excluded by reporting non-transfecting control. ii) The inhibition of sprouting by both gain and loss of function is surprising and is not explained by the proposed mechanism. iii) H-*Veal* is very lowly expressed in HUVEC (CT>31) making the siRNA experiment uninterpretable. iv) Finally, the functional assays raise some concerns: It is unclear how permeability can be enhanced in Figure 6J. Usually endothelial cells should be grown to confluence for this assay and permeability is induced by e.g. histamine.
4. The effects of *Veal2* on the distribution of PRKBC in Figures 4, Figure 6 and 7 is not convincing. No in vivo evidence that changes in localisation is affected and determines the mechanism of action is provided in vivo.

Specific comments (technical quality controls):

Figure 1C: please highlight regulated RNAs.

Figure 3C: the veal2 band in Prkcbb-IP is very weak. Fold change values may overestimate the data and raw data (CT or 2-CT values) should be shown.

Figure 3L: please report statistics on raw data instead of fold change

Figure 4F/G: the effect of veal2 on Prkcbb localization are minor (see above) and need to be statistically confirmed. Loading controls are missing.

Figure 5U: how does the ng (human Veal2) relate to the nM concentrations used for z-veal2. Please use nM also for this study and include the protein concentration in the figure legend to allow for an estimation of stoichiometry.

Since H-Veal appears very lowly expressed in HUVEC, dH2O controls should be shown and raw data should be shown for siRNA experiments (with controls).

Figure 7A/B: raw data should be shown in the figures instead of fold change

Figure 7D: please provide more details regarding blood samples in the figure legends. If endothelial exosomes are responsible for the increase this should be shown. Given that H-Veal2 is very lowly expressed in EC (at least in culture CT 30-31) it is difficult to reconcile that this should lead to a significant enrichment of exosomal RNA in the blood.

Referee #3:

This is an exciting and comprehensive study that identifies an endothelial-enriched lncRNA in zebrafish that binds to PRKCB2 to regulate its activity. Through genetic approaches in zebrafish they find that this lncRNA regulates vascular stability and that it interacts with PRKCB2. By pull down of PRKCB2 in human cells, they identify a human lncRNA that also regulates PRKCB2 by competing for DAG binding, although not sharing sequence identity. Furthermore, data is presented in human patients that demonstrates that this lncRNA may be important in regulating vascular permeability and may serve as a biomarker in diabetic retinopathy. For the most part the experiments are well controlled, the data is convincing, and the conclusions are based on supporting data. The findings will be of interest across multiple disciplines. However, there are several questions that remain.

Major comments:

- 1) It is not clear how abundant this lncRNA is in cells. How many copies per cell are there? While it is clear that knock-out or knock-down results in highly penetrant phenotypes, much of the mechanism is determined by performing activity assays of recombinant PRKCB2 in the presence of increasing amounts of the lncRNA. It is not clear if these experiments are physiologically relevant. It also isn't clear if the same amount of human and zebrafish lncRNA were added to the kinase assays. One is in nM and the other is in ng. Is there a cell-based assay that could be used to determine the effect of the lncRNA (and deletion mutants as controls) on PRKCB2 activity?
- 2) Is human VEAL2 also enriched in expression in endothelial cells?
- 3) Is the data presented in Fig. 4G significant? It looks like only a representative experiment. The figure legend indicates $n > 3$, but there are no error bars and no statistical test is indicated. Since this lncRNA is enriched in ECs, I don't understand how this result is possible with endothelial cells making up such a small proportion (<5%) of all cells in the total cell lysate. It would be helpful if there was a better way to assess Prkcb2 localization in zebrafish endothelium in vivo.
- 4) Were there any survival defects upon manipulation of z-veal2 or h-VEAL2 in human ECs? The matrigel tube formation results look very dramatic. What does the monolayer look like? Is it simply a matter of altered adherens junctions and thus permeability or is there also a cell survival

phenotype?

5) What is the effect of the diabetic conditions *in vitro* on h-VEAL2 expression?

Minor comments:

- 1) Does the zebrafish mutant with the 8 basepair deletion (nucleotides 26-33 of the transcript) still make normal levels of the lncRNA?
- 2) The naming convention is problematic since z-veal2 and h-VEAL2 are functional homologs, but share no sequence overlap or synteny. Is there a better naming convention that would make it clear that these are not true homologs.
- 3) Line 281 should be '*in vitro*' rather than '*in vivo*', since this is in cell culture cells.

LncRNA *VEAL2* regulates *PRKCB2* for modulating endothelial permeability and diabetic retinopathy

Response to reviewers

Referee #1:

The authors describe a pair of previously unstudied lncRNAs, in zebrafish and in human, that play important roles in the biology of endothelial cells. They also identify and report the mechanism of action of these lncRNAs, that bind and modulate the activity of the PRKCB protein involved in a signaling cascade important in the endothelial. Overall, the study is very well done and should be of high interest to both the lncRNA community and the endothelial cell community, and so fits well the scope of the journal. The data on zebrafish is overall solid and convincing. I do have some concerns about the human data, as detailed below.

Response) We appreciate the insightful comments of the reviewer. We have provided additional data as suggested by the reviewer and we have conducted experiments to address all the concerns. A pointwise response to the reviewer comments is as follows:

Major comment:

Comment 1A: *It is not clear from the existing and presented data how robustly/highly expressed is *VEAL2*. For the proposed activity it appears that it should be expressed at rather high levels. However, in ENCODE HUVEC RNA-seq data, there are barely any reads on the strand from which *VEAL2* is transcribed, and the FANTOM CAGE evidence for a promoter there is also very weak. In GTex data, it seems to be primarily expressed in the skin.*

Response 1A) We undertook fresh experiments to estimate the expression levels of *VEAL2* in HUVEC samples. We observed that *VEAL2* has an expression of ~25ct / 2µg of HUVEC input RNA. The absolute quantification of *VEAL2* in HUVECs showed that there are about 562 copies of *VEAL2* transcript per µg of RNA. The new data clearly revealed that *VEAL2* has a moderate expression in HUVECs. The previous observation on low expression of *VEAL2* could be attributed to very low amounts of RNA template as starting material. The new data is represented here and is given in figure no - **Expanded Figure 5H and Figure 5D** in the revised manuscript.

Expanded figure 5H: Bar graph representing relative expression of *VEAL2* in control HUVECs cells and hyperglycemia stimulated HUVECs by growing under high glucose. Data is acquired from 3 different biological replicates and shown as mean fold change values \pm standard deviation.

Figure 5D: Absolute quantification of *VEAL2* in HUVECs revealed 562 copies per μg of RNA. 10^2 - 10^8 copies of *VEAL2* RNA were used to make the standard curve.

Our observations for *VEAL2* transcript stem from a database on endothelial-specific transcriptome “ANGIOGENES” curated previously by Dr. Stefanie Dimmeler and group. ANGIOGENES database documented *VEAL2* as a known endothelial enriched candidate lncRNA. This database on endothelial cell transcriptome documented high expression of *VEAL2* (ENST00000413496) in different endothelial cell lines (Muller et al., 2016). This data is curated by ANGIOGENES database (<http://angiogenes.uni-frankfurt.de/transcript/ENST00000413496>) and is presented in the **Appendix Figure S 21A** of the revised manuscript and is also reproduced here.

Additionally, we have also conducted an experiment on expression analysis of *VEAL2* in primary cells originating from different human tissues. The q-RT-PCR based analysis clearly indicated *VEAL2* is enriched in human endothelial cells (HUVECs) over other primary cells such as Human Foreskin Fibroblast (HFF), Human Pancreatic Nestin Expressing cells (HPNE), Human Epidermal Melanocytes, neonatal, darkly pigmented (HEMn-DP), Primary Human Melanocytes, Peripheral Blood Mononuclear Cells (PBMC) and Human Hepatocyte Line 17 (HHL17). We have also noted that *VEAL2* originally identified by HAVANA project is not well represented in ENCODE data of HUVECs. We anticipate that with technical advancements and increase in RNA seq data coverage, such moderately expressed lncRNAs will be captured efficiently in future.

GTEx does not provide the expression data for *VEAL2* (ENSG00000232220). The transcript signal that is presented in GTEx is originating from another transcript ENSG00000232324, which originates from 5' end of MYADM and is expressed in the skin tissue. Our current experimental data and endothelial specific database concur with our findings of *VEAL2* expression in HUVECs.

Appendix Figure S 21A: Expression of *VEAL2* in different human primary endothelial cell lines curated by ANGIOGENES (<http://angiogenes.uni-frankfurt.de/transcript/ENST00000413496>). BEC (Brain endothelial cells), HAoEC (Human Aortic Endothelial Cells), HHSEC (Human Hepatic Sinusoidal Endothelial Cells), HMVEC-D (Human Dermal microvascular endothelial cells), HUVEC (Human umbilical vein endothelial cells).

Appendix Figure S 21B: *VEAL2* RNA levels are significantly high in endothelial cells (HUVECs) compared to other primary cells like Human Foreskin Fibroblast (HFF), Human Pancreatic Nestin Expressing cells (HPNE), Human Epidermal Melanocytes, neonatal, darkly pigmented (HEMn-DP), Primary Human Melanocytes, Peripheral Blood Mononuclear Cell (PBMC), Human Hepatocyte Line 17 (HHL17).

Comment 1B: Can the authors show RNA-seq tracks (from their RIP, and ideally from some strand-specific protocol) that show that the transcript is indeed originating from the antisense region of MYADM, and indeed spliced as in the gene model?

Response 1B) *VEAL2* is a known transcript identified earlier by HAVANA project and is cataloged as endothelial lncRNA by ANGIOGENES database. Our experimental data from 3'RACE conducted in HUVECs support these findings. We experimentally show that *VEAL2* as an anti-sense transcript in 3'UTR of MYADM gene. The 3' RACE data given in the figure below depict the direction of the transcript and its

spliced structure. We want to highlight that in all our experiments related to HUVECs or human patient data we have used primers (given in appendix table S5) designed on two exons targeting specifically spliced RNA. Though we are not able to differentially identify *VEAL2* from the ENCODE and RIPseq data, we were successful in providing experimental proof on the transcript structure and its origin. The data is presented in **Figure 5C** in the revised manuscript and is reproduced below.

Figure 5C: Representation of genomic location of AC008440.1- human vascular endothelial associated lncRNA 2 (*VEAL2*) transcript. *VEAL2* is antisense to 3' UTR of known protein coding myeloid-associated differentiation marker (*MYADM*) gene. 3'RACE experiment confirmed *VEAL2* as an independent transcript. Reads from PRKCB-RIPseq and IgG-RIPseq mapping to *VEAL2* loci are also given. Blue highlights reads mapping to +ve strand and Red highlights reads mapping to -ve strand.

Comment 1C: How abundant is it in HUVEC cells? These points are crucial to make the proposed functionality of *VEAL2* convincing.

Response 1C) As mentioned above in response to 1A, the recent data on absolute quantification clearly revealed that *VEAL2* has about 562 copies per μg of RNA indicating its moderate expression in HUVECs.

Minor comments:

Comment 1: Line 63, typo: "specialized epithelial cells that forms"

Response) Thank you for pointing out this error. We have corrected the error.

Comment 2: Lines 207-213, the part about Enzastaurin is confusing. The treatment by Enzastaurin does not cause any phenotype by itself? So the idea is that blocking the ATP-binding catalytic domain prevents *VEAL-2* from accessing it? This needs to be more clearly explained. There is also a typo in "Enz*u*staurin" in Fig. 3E.

Response) Enzastaurin is a known inhibitor for PRKCB2. We have treated zebrafish embryos lacking one copy of *veal2* (*veal2*^{gib005 Δ 8/+}) animals with enzastaurin.

In our experiments, *veal2*^{gib005Δ8/+} animals lack functional *veal2* mediated regulation of PRKCB2 in zebrafish, resulting in hyperactivity of *Prkcbb* in zebrafish leading to hemorrhage phenotype. To prove this further, we treated these *veal2*^{gib005Δ8/+} zebrafish embryos and showed that, by providing an external inhibitor, the hyperpermeability phenotype can be rescued. This complementation of *veal2* function by Enzastaurin in *veal2*^{gib005Δ8/+} animals confirmed that *veal2* negatively regulates zebrafish *Prkcbb* activity. This corroborates that *Prkcbb* hyperactivation is linked to hemorrhage phenotype in zebrafish. We have also rephrased the explanation given in line no 208-214 in the revised manuscript, as per suggestion. This is reproduced below:

“To confirm the inhibitory effect of veal2 on PRKCB2, we performed small molecule-based rescue in veal2^{gib005Δ8/+} animals using Enzastaurin. Enzastaurin is a chemical molecule known to block the ATP-binding catalytic domain of human PRKCB2. Enzastaurin treatment rescued hemorrhage phenotype in about 52% of offspring from the outcross of veal2^{gib005Δ8/+} animals compared to DMSO treatment (p-value<1E-4) (Figure-3F-K and 3L). This complementation of veal2 function by Enzastaurin in veal2^{gib005Δ8/+} animals confirmed that veal2 negatively regulates zebrafish Prkcbb activity.”

We also thank the reviewer for pointing out the typo in figure 3E, a corrected image has been provided in the revised manuscript.

Comment 3: Line 223: Does mutation of motif-3 affect the ability of VEAL-2 to inhibit PRKCB *in vitro*?

Response) In our original manuscript we already had identified and queried the activity of all 4 putative motifs in *veal2* in the *in-vivo* system. In the previous version of the manuscript, we had not tested an *in-vitro* effect of motif 3 in *veal2* on *Prkcbb* activity. Since this same clarification is requested by other reviewers too, we have conducted fresh experiments to determine the effect of all 4 motifs in *veal2* on the *in-vitro* kinase activity of PRKCB2. The *veal2* variants lacking motif 1,2,4 independently did not alter the inhibitory function of *veal2* on PRKCB2 kinase activity. Whereas, *veal2* lacking motif 3 (also called as C1 domain binding LncRNA Motif (CLM)) was unable to inhibit kinase activity indicating its functional significance. This data recapitulates the findings observed *in-vivo* zebrafish model. The data is provided in **Appendix Figure S 20C** in the revised manuscript and is represented below:

S20 C

Appendix Figure S 20C: Relative kinase activity of human PRKCB2 in the presence of various variants of *veal2* lacking putative functional motifs and wildtype *veal2* IVT RNA. Data from three different experiments plotted as mean fold change values \pm standard deviation.

Comment 4: Line 281: Please include a KD control for the FISH of VEAL2 (in light of the concerns above)

Response) We agree with the reviewer that FISH for *VEAL2* in “*VEAL2* knockdown (KD) HUVEC cells” will highlight the specificity of the assay. Therefore we transfected HUVECs with siRNA targeting *VEAL2* and performed the FISH. We observed an absence of *VEAL2* signal in HUVECs transfected with siRNA targeting *VEAL2*. This data is represented below and is given in the revised manuscript as **Figure 6F and Expanded View Figure 3E-F.**

Expanded view figure 3E-F: smFISH of *VEAL2* in HUVECs transfected with control siRNA and *VEAL2* siRNA shows specificity of cytoplasmic signal of *VEAL2*. *VEAL2* in CAL-fluor red(610nM). E-F. Merged image for *VEAL2* and DAPI. Magnification-100X, scale bar-5 μ m.

Figure 6F. siRNA mediated knockdown of *VEAL2* significantly reduces expression of *VEAL2* in HUVECs.

Referee-#2:

The present study reports a novel endothelial expressed lncRNA, named veal2, which control vascular permeability in zebrafish. The authors further report that veal2 interact with Prkcbb to limit its enzymatic activity. Pharmacological inhibition of Prkcbb prevented veal2 mutant phenotypes in zebrafish. The strength of the study is that authors report a phenotype of the veal2 mutant which is rescued by zebrafish and human Veal2. However, the study has several weakness as outline below.

Response) We appreciate the constructive comments of the reviewer. We have provided answers to all the concerns raised by the reviewer. A pointwise response to the reviewer comments is as follows:

***Comment 1A:** It is unclear why the 8 bp deletion was chosen as zveal2 loss of function model. In the follow up studies, the authors identified motif 3 as putative interaction motif. If this is true, motif 3 mutations should be studied in zebrafish.*

Response) In our study, we used a reverse genetics approach to understand the role of lncRNA veal2 in regulating endothelial cell function. Initially, we targeted veal2 using two pairs of TALEN to delete the whole lncRNA in zebrafish. But the deletion of whole lncRNA was lethal in zebrafish (Appendix Figure S 6). Later we targeted 5' and 3' end of veal2 loci independently and observed that indels at 5' end in veal2 loci cause endothelial-specific phenotype. The 8 bp deletion in 5' end of veal2 loci was chosen randomly based on high fecundity and health of founder zebrafish animals with hemorrhage phenotype. Later using *in-vitro* and *in-vivo* assays, we identified Prkcb2 as an interacting partner for veal2.

To probe the specific binding sites between veal2 and Prkcbb, we conducted *in-silico* simulations studies. We identified motif3 in veal2 as the probable site for interaction with Prkcb2. The role of motif3 in veal2 was validated using complementation assay in zebrafish. Using classical genetic techniques, we synthesized and supplemented the veal2 variant lacking motif3 in veal2^{gib005Δ8/+} embryos and found that it is unable to rescue the hemorrhage phenotype compared to full-length wild type veal2 RNA. Therefore we have studied the specific effect of motif3 in veal2 in zebrafish embryos by performing complementation experiments. This data is represented in **Figure 4C** and is reproduced here.

Further, to identify the effect of the 8 bp deletion in veal2 (veal2-Δ8) on motif3 in veal2, we conducted similar *in-silico* simulations of veal2-Δ8 RNA with Prkcb2. This revealed that different structural conformation of veal2-Δ8 alters the RNA folding,

preventing the interaction between motif3 of *veal2* and Prkcbb (**Figure 4B**). This indicates that the 8bp deletion changed the structural conformation of *veal2* and resulted in a loss of interaction between motif3 in *veal2* and Prkcb2.

Figure 4: A-B. Enlarged view of the site of interaction of *veal2* WT and mutant *veal2*- $\Delta 8$ RNA with Prkcbb protein. Four motifs were predicted in *veal2* interacting with Prkcbb, highlighted in different colors. Motif1- (549, 551- 553, 559, 560) in Purple, motif2- (584- 587) in Orange, motif3 (678- 680) in Red, motif4- (844- 846) in Green. The deletion in *veal2*- $\Delta 8$ RNA led to change in folding and altered interaction with Prkcbb.

C. Scatter plot showing the number of animals that displayed the hemorrhage phenotype across the progeny of an outcross of *veal2*^{gib005 Δ 8/+} zebrafish injected with IVTs of different *veal2* variants. Control indicates *veal2*^{gib005 Δ 8/+} zebrafish without complementation. Data from three different experiments plotted as mean percentage values \pm standard deviation.

Comment 1B: In addition, the biochemistry supporting Figure 4 is weak and the relevance of the interaction motifs should be further supported by *in vitro* experiments showing the effects of *veal2* deletion/point mutations on enzyme activity. Adding human *Veal2* / mutants would also help to confirm that the human *lncRNA*, which according to the authors lacks sequence and structural homology, is indeed a functional orthologue.

Response) In our original manuscript we had identified and queried the functions of four independent putative motifs of *veal2* in the *in-vivo* zebrafish model. This revealed motif 3 in *veal2* as a functional interaction site with PRKCB2. Further a *veal2* variant lacking motif-3 was unable to rescue the hemorrhage phenotype in *veal2*^{gib005Δ8/+} embryos.

In addition to this as suggested by the reviewer, we further probed the effect of the four putative motifs of zebrafish *veal2* using an *in-vitro* kinase assay. The zebrafish *veal2* variants lacking motif 1, 2 & 4 independently could not alter the inhibitory function of *veal2* on PRKCB2 kinase activity. Whereas, *veal2* lacking motif 3 (also called as C1 domain binding LncRNA Motif (CLM)) was unable to inhibit kinase activity indicating its functional significance. This data recapitulate the findings observed previously using the *in-vivo* zebrafish model.

Next we also tested the impact of CLM motif identified in human *VEAL2*. The human *VEAL2* variant lacking CLM motif (*VEAL2* ΔCLM) were unable to regulate kinase activity of PRKCB2 in an *in-vitro* assay. This indicated that both zebrafish *veal2* and human *VEAL2* interact through the “C1 binding domain” in the PRKCB2 protein and this interaction is required for regulation of PRKCB2 kinase activity. The data is shown in **Appendix Figure S 20C and EV 2D** and is represented below.

S20 C

EV 2D

Appendix Figure S 20C: Relative kinase activity of human PRKCB2 in the presence of various variants of zebrafish *veal2* lacking putative functional motifs and wildtype *veal2* IVT RNA. Data from three different experiments plotted as mean fold change values ± standard deviation.

Expanded figure 2D: Relative kinase activity of human PRKCB2 in the presence of wildtype human *VEAL2* RNA and human *VEAL2* variant RNA lacking putative CLM motif (*VEAL2* Δ CLM). Data from three different experiments plotted as mean fold change values \pm standard deviation.

Comment 2: *The effects of veal2 on zebrafish and human EC function are not fully compatible since endothelial cells in culture show a profound anti-angiogenic response (likely also toxicity), which is not observed to that extent in zebrafish. This might be in part related to the lack of conservation or vascular bed specific actions of Veal2. It may help to provide a more detailed analysis of vessel morphology (including high magnification images) in the zebrafish.*

Response) We have used zebrafish as an *in-vivo* system to screen functional lncRNAs in endothelial cell function at the organismal level. Later we used HUVECs to unveil the molecular mechanism of *VEAL2* mediated regulation of PRKCB2 in endothelial cells. Each system has its own limitations and advantages. In our study, editing of *veal2* loci in zebrafish resulted in endothelial dysfunction and blood vessel integrity defect causing hemorrhage. In literature there are many other reports that highlighted hemorrhage phenotype in various endothelial dysfunctions in zebrafish (Buchner et al., 2007, Fish et al., 2008, Kwon et al., 2012, Lalwani et al., 2012). During early zebrafish embryonic stages, tissue specific vasculature is still under active development and is dynamic in nature, hence vascular bed specific phenotyping may not be feasible in zebrafish embryonic stages. However in our study, using confocal microscopy based high-resolution imaging, we observed lack of cerebral Central arteries (CtAs) in *veal2*^{gib005 Δ 8/+} embryos as presented in **Figure 2N (arrowhead indicate lack of CtAs)**. The loss of CtAs led to the formation of hemorrhage and it was used as a surrogate phenotype. As suggested by the reviewer, we planned to conduct live imaging of zebrafish embryos to better document the vessel morphology. In order to undertake this experiment the lead author(s) and the zebrafish embryos would have to physically travel to a collaborator's laboratory in a different city. This was not possible due to strict travel restrictions imposed on account of COVID19 pandemic.

Figure 2 I-O: Representative images of 2 dpf zebrafish which displayed vascular integrity defects in the progeny of an outcross of *veal2gib005Δ8/+* zebrafish. **I-J.** Control zebrafish embryos. **N-O.** *veal2gib005Δ8/+* embryos.

We agree with the reviewer that we don't observe a similar severe phenotype in zebrafish as observed in HUVEC studies. In HUVECs we observed a severe anti-angiogenesis effect. Both zebrafish and HUVEC are unique systems, we don't intend to observe exactly same phenotypes. This is documented earlier in the literature as well. Among many reports, the loss of function of *DYRK1A* and *RAP1B* in zebrafish caused blood vessel integrity defects leading to hemorrhage without any severe angiogenesis defects (Cho et al., 2019, Gore et al., 2008). Whereas siRNA-mediated knockdown of *DYRK1A* and *RAP1B* in HUVECs caused a severe decline in angiogenesis (Figure 6B in Rozen et al., 2018, Figure 4 in Yan et al., 2008).

To further delineate the toxicity effect in HUVECs, we conducted an apoptosis assay in HUVECs. The Lactate dehydrogenase (LDH) based detection of apoptosis in HUVECs under overexpression and knockdown of *VEAL2* conditions showed a lack of any cell toxicity signals. This gave more clarity and we have incorporated this data in the revised manuscript. The data is given in **Appendix Figure S 22D** and is also reproduced here.

S22 D

Appendix Figure S 22D: Dot plot representing lactate dehydrogenase levels measured in HUVECs transfected with different overexpression plasmids or siRNAs. Data is acquired from 3 different biological replicates and shown as mean fold change values \pm standard deviation.

Comment 3: In addition, the *in vitro* studies are not very convincing and have several technical and conceptual weaknesses:

i) the control plasmid and control siRNA transfected cells show major differences in baseline values. Unspecific effects of the transfection protocols should be taken into account and excluded by reporting non-transfecting control.

Response) We want to highlight here that in both overexpression and knockdown assays, we have used the same nucleofection (Lonza) based transfection reagents. The background toxicity due to transfection protocol will be similar in both experiments. However, size differences of control plasmid and non-targeting siRNA control may explain the differences in the toxicity levels observed. In summary, both sets of experiments had different experimental controls (plasmid vs siRNA). Further, as per the reviewer's suggestion, we have included non-transfection control in our study in the revised manuscript. We studied the junction formation of HUVECs in non-transfection conditions. Although the number of junctions in non-transfection controls was high (~100) compared to control plasmid (~40) and *VEAL2* overexpression (~5), still the change in the number of junctions was significantly less upon *VEAL2* overexpression. This supplementary information is presented in Appendix Figure S S22 in revised manuscript and is reproduced here.

ii) The inhibition of sprouting by both gain and loss of function is surprising and is not explained by the proposed mechanism.

Response) PRKCB2 has also been shown earlier to control angiogenesis (Nakamura et al.,2010). The substrate of PRKCB2, a known potent endothelial junctional protein VE-cadherin (CDH5) has been shown to regulate angiogenesis as well. It is widely documented that CDH5 expression on membrane control integrity and plasticity of endothelial cells (Harris et al., 2010, Szymborska et al.,2017). The loss of CDH5 has been shown to cause an anti-angiogenesis effect (Carmeliet et al.,1999). Whereas, increase in CDH5 expression also caused reduced proliferation and angiogenesis via regulating VEGFR2 (Harris et al., 2010). Hence CDH5's optimal levels are essential to maintain proper endothelial cell function. Additionally, there are many more examples in the literature where both loss and gain of function of proteins lead to similar phenotype due to perturbation in their modulatory activity (Zylinska et al., 2009, Bernick et al.,2010, Wang et al.,2014,). Here in this study, we have majorly focused on *VEAL2* mediated regulation of PRKCB2 control over junctional dynamics involved in maintaining permeability. Although the effect on angiogenesis is an important observation, our study is largely limited to understanding how *VEAL2* mediated regulation of PRKCB2 affects junctional dynamics in endothelial cells. However in future we look forward to investigating the specific role of *VEAL2* in angiogenesis.

iii) *H-Veal* is very lowly expressed in HUVEC (CT>31) making the siRNA experiment uninterpretable.

Response) We undertook fresh experiments to estimate the expression levels of *VEAL2* in HUVEC samples. We observed that *VEAL2* has an expression of ~25ct / 2µg of HUVEC input RNA. The absolute quantification of *VEAL2* in HUVECs showed that there are about 562 copies of *VEAL2* transcript per µg of RNA. The new data clearly revealed that *VEAL2* has moderate expression in HUVECs. The previous observation on low expression of *VEAL2* could be attributed to low amounts of RNA template as starting material. The new data is represented here and is given in figure no - **Figure 5D** in the revised manuscript.

On the other hand, we had already validated the effect of siRNAs in our experiments. Both relative quantifications presented in figure 6F and freshly conducted FISH data as given below (provided in the Expanded figure 3E-F in revised manuscript) clearly show the effective knockdown of *VEAL2* in HUVECs. The data is given here below.

Figure 5D: Absolute quantification of *VEAL2* in HUVECs revealed 562 copies expressed per µg of RNA. 102-108 copies of *VEAL2* RNA were used to make the standard curve.

Expanded figure 3E-F: smFISH of *VEAL2* in HUVECs transfected with control siRNA and *VEAL2* siRNA shows specificity of cytoplasmic signal of *VEAL2*. *VEAL2* in

CAL-fluor red(610nM). E-F. Merged image for *VEAL2* and DAPI. Magnification-100X, scale bar-5µm.

Figure 6F. siRNA mediated knockdown of *VEAL2* significantly reduces expression of *VEAL2* in HUVECs.

iv) Finally, the functional assays raise some concerns: It is unclear how permeability can be enhanced in Figure 6J. Usually endothelial cells should be grown to confluence for this assay and permeability is induced by e.g. histamine.

Response) As per our findings, *VEAL2* regulates the activity of PRKCB2 to maintain junctional dynamics in endothelial cells. In the case of *VEAL2* knockdown such inhibitory handle on PRKCB2 is lacking, leading to hyperactivation of PRKCB2 and altered junctional dynamics to cause hyperpermeability. It has been observed previously that treatment of HUVECs with phorbol myristate acetate (PMA), without any induction, causes an increase in permeability in HUVECs (Fig 7 in Waschke et al.,2006). PMA is a known activator for PRKCB2, hence our findings also are concordant with the literature. It indicates that hyperactivation of PRKCB2 by PMA or *VEAL2* without any induction can cause hyperpermeability in endothelial cells.

Comment 4: *The effects of Veal2 on the distribution of PRKCB in Figures 4, Figure 6 and 7 is not convincing. No in vivo evidence that changes in localisation is affected and determines the mechanism of action is provided in vivo.*

Response) The membrane localization of Prkcbb/PRKCB2 is a highly dynamic process and PRKCB2 is abundantly located in the cytoplasm at a given time. Visually tracking the localization of Prkcbb in zebrafish is difficult due to the non-availability of specific antibodies. Measuring kinase activity of PRKCB2 or tracking the assembly/localization of its substrates i.e. junctional proteins on the membrane is a better readout to determine the activity of PRKCB2 in cell milieu. Hence we have used the “kinase activity of PRKCB2” and “tracking of the assembly/localization of the substrates” as surrogate readouts for indicating PRKCB localization in cells. We conducted the following experiments in our manuscript to show *VEAL2* mediated regulation of PRKCB2 activity.

- i. In our manuscript, we utilized phenotypic readout of hyperpermeability in the form of hemorrhage in zebrafish. The lack of inhibitory effect of *veal2* on Prkcbb in *veal2*^{gib005Δ8/+} embryos caused hemorrhage phenotype. The supplementation of zebrafish *veal2* or human *VEAL2*, or a known inhibitor of

Prkccb, Enzastaurin, rescued the hyperpermeability phenotype. These *in-vivo* experiments indicated *veal2* mediated control of Prkccb activity.

- ii. In HUVECs, we detected both localization of PRKCB2, and its substrates (CDH5 and CTNNB1) and hyperpermeability phenotype. Translocation of CDH5 and CTNNB1 signals to the cytoplasm in HUVECs transfected with siRNA targeting human *VEAL2* clearly provides evidence that PRKCB2 is hyperactive in the absence of *VEAL2*. In literature, it is well documented that upon hyperactivation, PRKCB2 mobilizes to the membrane and phosphorylates junctional proteins (Haidari et al., 2014, Cardiovasc Diabetol.). Hence our findings support documented reports and identify a novel regulator of PRKCB2.

To support these findings further and to get a better readout from *in-vivo* models, we conducted an additional experiment, in which we measured kinase activity of PRKCB in *veal2*^{gib005Δ8/+} embryos and different HUVEC models. The cell lysate from the above-mentioned samples was used to measure endogenous kinase activity of Prkccb/PRKCB2 using *in vivo* kinase activity assay. The inhibition in the endogenous activity of PRKCB2 in HUVECs, in presence of zebrafish *veal2* or human *VEAL2*, confirmed lncRNA-based modulation of PRKCB2 activity in an *in-vivo* system (**Expanded view figure 4A-B**). In summary, the hyper permeability phenotype in zebrafish and HUVECs, impact of zebrafish *veal2* and human *VEAL2* on endogenous kinase activity of Prkccb/PRKCB2, in zebrafish model and HUVECs and localization of PRKCB2 and its substrate junctional proteins in HUVECs clearly underscore that *VEAL2* regulates localization and hence activity of PRKCB2.

Expanded figure 4A: Dot plot representing endogenous kinase activity of human PRKCB2 in HUVECs under standard conditions and upon overexpression of wildtype zebrafish *veal2*, various variants of zebrafish *veal2*, human *VEAL2*, and its variant. Data from three different experiments plotted as mean fold change values \pm standard deviation.

Expanded figure 4B: Dot plot representing endogenous kinase activity of human PRKCB2 in HUVECs under standard conditions and upon knockdown of human *VEAL2*. Data from three different experiments plotted as mean fold change values \pm standard deviation.

Specific comments (technical quality controls):

Comment 1: Figure 1C: please highlight regulated RNAs.

Response) We have made appropriate changes as per suggestions.

Comment 2: Figure 3C: the *veal2* band in *Prkcbb*-IP is very weak. Fold change values may overestimate the data and raw data (CT or 2-CT values) should be shown.

Response) We have modified the data representation. This is presented as figure 3D in the revised manuscript and is reproduced below.

Figure 3D: qRT-PCR based quantification of *veal2* transcript across *Prkcbb*, *Ephb3* and IgG immunoprecipitations.

Comment 3: Figure 3L: please report statistics on raw data instead of fold change

Response) Statistics are applied to raw data only. We computed an unpaired T-test on raw phenotypic numbers given in percentage. This data is already given in appendix table S 12 (supporting data for figure 3). We have reproduced the same data here:

Raw data	
DMSO	Enzastaurin 10 μ M
53.3	27.72
50.4	26.12
51.36	23.47

Statistics	
Unpaired t test	
P value	<0.0001
P value summary	****
Significantly different (P < 0.05)?	Yes
One- or two-tailed P value?	Two-tailed
t, df	t=17.23, df=4

Comment 4: Figure 4F/G: the effect of *veal2* on *Prkcb* localization are minor (see above) and need to be statistically confirmed. Loading controls are missing.

Response) We have provided this information in the revised manuscript. The membrane localization of *Prkcb*/*PRKB2* is a highly dynamic process and is abundantly located in the cytoplasm at a given time under normal conditions. In our present study, we also observed a similar dynamic effect in our data. The statistics for all 3 replicates are given in appendix Table S no. 13.

Figure 4 F-G. Abundance of *Prkcb* protein is significantly enriched on membrane in *veal2^{gib005Δ8/+}* embryos compared to control. T- Total, C- Cytoplasmic & M- Membrane.

Comment 5: Figure 5U: how does the ng (human *Veal2*) relate to the nM concentrations used for *veal2*. Please use nM also for this study and include the protein concentration in the figure legend to allow for an estimation of stoichiometry.

Response: We apologize for this mistake. We have provided the correct figure in the revised manuscript. We have also added information on the concentration of PRKCB2 used in the assay in both legend and axis label. The data is reproduced below.

Figure 3E. Relative kinase activity of human PRKCB2 under standard conditions and in the presence of various concentrations of the wildtype zebrafish *veal2* RNA, *veal2*- Δ 8 RNA, *veal2*-AS RNA. 52nM of the PRKCB2 protein was used per reaction.

Figure 5U. Relative kinase activity of human PRKCB2 under standard conditions and in the presence of various concentrations of the wildtype human *VEAL2* RNA and *VEAL2*-AS RNA. 52nM of the PRKCB2 protein was used per reaction.

Comment 6: Since H-Veal appears very lowly expressed in HUVEC, dH2O controls should be shown and raw data should be shown for siRNA experiments (with controls).

Response) The re-estimation of *VEAL2* expression in a higher concentration of HUVEC RNA indicated that *VEAL2* is moderately expressed lncRNA. The absolute quantification also showed that about 562 copies/ μ g of *VEAL2* is expressed in HUVECs. Additionally, we have also provided dH2O (negative control) data. No Ct or Ct>36 in negative control indicate specificity of *VEAL2* signals in HUVECs. The raw data is provided in **appendix table S no-14**.

Comment 7: Figure 7A/B: raw data should be shown in the figures instead of fold change

Response) We have made the suggested changes in data representation in our revised manuscript. These are also reproduced here:

Figure 7A: Increased ΔCt values $VEAL2$ levels highlighting reduced expression of $VEAL2$ in choroid tissue isolated from diabetic retinopathy (DR) patients compared to controls, indicating its r. N=8.

Figure 7B: Reduced ΔCt values of $VEAL2$ in blood samples of diabetic patients highlighting significantly higher expression in diabetic conditions compared to controls. Direct correlation of increased $VEAL2$ expression was seen in blood samples of patients with different diabetic stages with aggravating vascular dysfunctions from diabetes mellitus (DM) to non-proliferative diabetic retinopathy (NPDR) to proliferative diabetic retinopathy (PDR) compared to control patients. N=50 each.

Comment 8A: Figure 7D: please provide more details regarding blood samples in the figure legends.

Response) Figure 7D is related to the hyperglycemia model created in HUVECS and its amelioration by overexpressing zebrafish *veal2* or human *VEAL2*.

Details of the blood samples representing the disease conditions for Figures 7B and 7C are given in the material section.

Comment 8B: If endothelial exosomes are responsible for the increase this should be shown. Given that *VEAL2* is very lowly expressed in EC (at least in culture CT 30-31) it is difficult to reconcile that this should lead to a significant enrichment of exosomal RNA in the blood.

Response) Endothelial cells are known to secrete exosomes to regulate exocrine signaling (Davidson et al., 2018, Guo et al., 2020). We wanted to find out if *VEAL2* is expressed in exosomes secreted by endothelial cells. In published literature by Perez-Boja et al (2018), we noticed that ENST00000413496 (*VEAL2*) levels are elevated in exosomes released by endothelial cells. So we speculate that

hyperglycemia-induced exosomes from ECs could be a source of increased *VEAL2* level in the blood to communicate with other tissues/cells. We agree with the reviewer that lowly expressed lncRNA may not lead to significant enrichment in exosomes, but our recent data incorporated in the revised manuscript (see response to comment 3(iii)) showed that *VEAL2* has moderate expression in HUVECs. It will be interesting to study the role of exosomes mediated signalling of *VEAL2* in hyperglycemia in the future.

Referee #3:

This is an exciting and comprehensive study that identifies an endothelial-enriched lncRNA in zebrafish that binds to PRKCB2 to regulate its activity. Through genetic approaches in zebrafish they find that this lncRNA regulates vascular stability and that it interacts with PRKCB2. By pull down of PRKCB2 in human cells, they identify a human lncRNA that also regulates PRKCB2 by competing for DAG binding, although not sharing sequence identity. Furthermore, data is presented in human patients that demonstrates that this lncRNA may be important in regulating vascular permeability and may serve as a biomarker in diabetic retinopathy. For the most part the experiments are well controlled, the data is convincing, and the conclusions are based on supporting data. The findings will be of interest across multiple disciplines. However, there are several questions that remain.

Response) We really appreciate the positive feedback from the reviewer. We have conducted various experiments for answering all the suggested experiments. The point-wise response to the reviewer comments is as follows:

Major comments:

Comment 1: *It is not clear how abundant this lncRNA is in cells. How many copies per cell are there? While it is clear that knock-out or knock-down results in highly penetrant phenotypes, much of the mechanism is determined by performing activity assays of recombinant PRKCB2 in the presence of increasing amounts of the lncRNA. It is not clear if these experiments are physiologically relevant. It also isn't clear if the same amount of human and zebrafish lncRNA were added to the kinase assays. One is in nM and the other is in ng. Is there a cell-based assay that could be used to determine the effect of the lncRNA (and deletion mutants as controls) on PRKCB2 activity?*

Response) We thank the reviewer for this suggestion. We have included information on the copy number of VEAL2 in HUVECs in our revised manuscript. The absolute quantification of VEAL2 in HUVECs indicated VEAL2 is a moderately expressed lncRNA and has 562 copies per μg of RNA. We also re-quantified the relative expression of VEAL2 in HUVECs using a higher amount of RNA. We detected that VEAL2 has $\sim 25\text{ct}/2\mu\text{g}$ of RNA. These recent data clearly explain that VEAL2 is moderately expressed lncRNA. The new data is represented here and is given in **Figure 5D** in the revised manuscript.

5D

Figure 5D: Absolute quantification of *VEAL2* in HUVECs revealed 562 copies expressed per μg of RNA. 102-108 copies of *VEAL2* RNA were used to make the standard curve.

We apologize here for wrongly using two different units in a similar experiment. We have made the required changes in the revised manuscript.

Figure 3E. Relative kinase activity of human PRKCB2 under standard conditions and in the presence of various concentrations of the wildtype zebrafish *veal2* RNA, *veal2- Δ 8* RNA, *veal2-AS* RNA. 52nM of the PRKCB2 protein was used per reaction.

Figure 5U. Relative kinase activity of human PRKCB2 under standard conditions and in the presence of various concentrations of the wildtype human *VEAL2* RNA and *VEAL2-AS* RNA. 52nM of the PRKCB2 protein was used per reaction.

In response to queries raised by reviewers for cell based assay, we conducted an additional experiment to measure the *in-cell* kinase activity of PRKCB2. We measured the endogenous kinase activity of PRKCB2 in different HUVEC conditions. The cell lysate from HUVECs transfected with different zebrafish *veal2* and human *VEAL2* variants were used to measure kinase activity using a PKC Kinase Activity

Assay Kit (ab139437, Abcam, UK). This data underscores that wildtype zebrafish *veal2* and human *VEAL2* can inhibit the *in-cell* endogenous activity of PRKCB2 in concentration-dependent manner (**Expanded figure 4A**). Various deletion mutants including zebrafish *veal2* lacking 8bp (*veal2* Δ 8), zebrafish *veal2* lacking motif3 (CLM) (*veal2* Δ CLM), human *VEAL2* lacking CLM (*VEAL2* Δ CLM) had no effect on PRKCB2 kinase activity, highlighting their potential functional relevance. We also tested *in-cell* kinase activity in the *VEAL2* knockdown model in HUVECs and found that endogenous kinase activity is relatively higher in HUVECs transfected with siRNA targeting human *VEAL2* compared to control siRNA transfected cells (**Expanded view figure 4B**). The new data is represented here.

EV 4A

EV 4B

Expanded figure 4A: Dot plot representing endogenous kinase activity of human PRKCB2 in HUVECs under standard conditions and upon overexpression of various variants of zebrafish *veal2* lacking 8bp (*veal2* Δ 8), zebrafish *veal2* lacking motif3 (CLM), (*veal2* Δ CLM), human *VEAL2* lacking CLM (*VEAL2* Δ CLM) and its variant. Data from three different experiments plotted as mean fold change values \pm standard deviation.

Expanded figure 4B: Dot plot representing endogenous kinase activity of human PRKCB2 in HUVECs under standard conditions and upon knockdown of *VEAL2*. Data from three different experiments plotted as mean fold change values \pm standard deviation.

Comment 2: *Is human VEAL2 also enriched in expression in endothelial cells?*

Response) In our study, we have identified and characterized human *VEAL2* in human endothelial cells (HUVECs). Our observation on *VEAL2*'s endothelial specificity stems from ANGIOGENES database by Prof Dimmeler's group, which has catalogued *VEAL2* (ENST00000413496) as an endothelial-specific lncRNA (Muller et al., 2016). They identified high expression of *VEAL2* in different endothelial cell lines. This data is given below and is presented in **Appendix Figure S 21A**.

S21 A

Appendix Figure S 21A: Expression of *VEAL2* in different human primary endothelial cell lines curated by ANGIOGENES (<http://angiogenes.uni-frankfurt.de/transcript/ENST00000413496>) BEC(Brain endothelial cells), HAoEC (Human Aortic Endothelial Cells), HHSEC (Human Hepatic Sinusoidal Endothelial Cells), HMVEC-D (Human Dermal microvascular endothelial cells), HUVEC (Human umbilical vein endothelial cells).

Further, we also performed an additional expression analysis of *VEAL2* in primary human cells originating from different tissues using q-RT-PCR. This data clearly indicates that *VEAL2* expression is enriched in HUVECs among other queried primary human cell lines such as Human Foreskin Fibroblast (HFF), Human Pancreatic Nestin Expressing cells (HPNE), Human Epidermal Melanocytes, neonatal, darkly pigmented (HEMn-DP), primary human melanocytes, peripheral blood mononuclear cells (PBMC) and Human Hepatocyte Line 17 (HHL17). The new data is incorporated in the revised manuscript and represented here.

S21 B

Appendix Figure S 21B: *VEAL2* RNA levels significantly high in endothelial cells (HUVECs) compared to other primary cells like foreskin fibroblast (HFF), Human Pancreatic Nestin Expressing cells (HPNE), Human Epidermal Melanocytes Neonatal Darkly Pigmented (HEMn-DP), Primary Human Melanocytes, Peripheral Blood Mononuclear Cell (PBMC), Human Hepatocyte Line 17 (HHL17).

Comment 3: Is the data presented in Fig. 4G significant? It looks like only a representative experiment. The figure legend indicates $n > 3$, but there are no error bars and no statistical test is indicated. Since this lncRNA is enriched in ECs, I don't understand how this result is possible with endothelial cells making up such a small proportion (<5%) of all cells in the total cell lysate. It would be helpful if there was a better way to assess *Prkcb2* localization in zebrafish endothelium *in vivo*.

Response) In our original manuscript we had conducted 3 biological replicates using approx 800 zebrafish embryos in each replicate. However, only one replicate was given here as representation. We have provided data for all three replicates including loading controls in revised manuscript. The data is presented in figure 4F-G and is given below.

Figure 4F-G. Abundance of Prkcbb protein is significantly enriched on membrane in *veal2*^{gib005Δ8/+} embryos compared to control. T- Total, C- Cytoplasmic & M-Membrane.

We assayed the differential localization of Prkcbb in control and *veal2*^{gib005Δ8/+} embryos. A large number (5,000 to 8,000) of transgenic zebrafish embryos are required for FACS followed by subcellular fractionation for checking expression specifically in endothelial cells. Sometimes the zebrafish do not produce such a large number of embryos on a daily basis due to subtle changes in environmental and housing conditions. When this experiment was scheduled a large number of zebrafish embryos were not readily available. Therefore we used whole-cell lysate instead of FACS-sorted ECs as a practical alternative, knowing that *veal2* is an endothelial-specific lncRNA.

The membrane localization of Prkcbb/PRKB2 is a highly dynamic process and is abundantly located in the cytoplasm at a given time. Many available antibodies for zebrafish are not immunofluorescence grade. We tried documenting live tracking of

Prkcbb in zebrafish, but without any success. Therefore, we measured the activity of Prkcbb/PRKCB2 in the form of junctional assembly on membrane and permeability levels as final readout. We utilized zebrafish as an *in-vivo* system to measure endpoint permeability defects in *veal2*^{gib005Δ8/+} embryos and subtle increased localization of Prkcbb on membrane by performing twestern-based assay.

To support these findings, we analyzed the visual localization of PRKCB2, its substrates (CDH5 and CTNNB1) via immunofluorescence in HUVECs and by measuring permeability levels in the HUVEC monolayer. The aberrant expression of CDH5 and CTNNB1 is evident in HUVECs transfected with siRNA targeting human *VEAL2* presented in figure 6I-T.

To further support this finding we conducted an additional experiment to determine the *in-cell* (endogenous) kinase activity of Prkcb2 in *veal2*^{gib005Δ8/+} embryos by measuring endogenous kinase activity. The elevated kinase activity of Prkcbb in *veal2*^{gib005Δ8/+} embryos indicates an alteration in its endogenous activity highlighting the imperative role of *veal2* in regulating its function. This data is presented in **Appendix Figure S 17C** and is given below. Taken together, i) the hyper permeability phenotype in zebrafish and HUVECs; ii) impact of *veal2/VEAL2* on endogenous kinase activity of Prkcbb/PRKCB2 *in zebrafish* and HUVEC model; and iii) expression analysis of PRKCB2 and its substrate junctional proteins in HUVECs clearly demonstrate that *VEAL2* regulates localization and activity of PRKCB2 under *in-vivo* conditions.

Appendix Figure S 17C: Dot plot represents *in-vivo* endogenous kinase activity of Prkcbb in control and in *veal2*^{gib005Δ8/+} animals. Data from three different experiments plotted as mean fold change values ± standard deviation.

To support these findings, we further analyzed the visual localization of PRKCB2, its substrates (CDH5 and CTNNB1) via immunofluorescence in HUVECs and by measuring permeability levels in the HUVEC monolayer. The aberrant expression of CDH5 and CTNNB1 is evident in HUVECs transfected with siRNA targeting human *VEAL2*. This data is presented in **Figure 6**. Therefore by using CDH5 and CTNNB1 (substrates of PRKCB2) as readouts we were able to assess the localization of PRKCB2 in HUVECs.

Taken together we demonstrate using two independent models, namely zebrafish and HUVECs, that i) Prkcbb protein is significantly enriched on endothelial cell membrane in zebrafish embryos that lack a copy of *veal2*; ii) Since *veal2* is an endothelial-specific lncRNA, the changes in membrane localization of Prkcbb is restricted to endothelial cells; iii) Elevated *in-cell* kinase activity of Prkcbb was noticed in zebrafish embryos lacking a copy of *veal2* indicating the role of *veal2* in regulating Prkcbb function in zebrafish embryos; iv) impact of *veal2/VEAL2* on endogenous kinase activity of Prkcbb/PRKCB2; and v) expression analysis of PRKCB2 and its substrate junctional proteins (CDH5 and CTNNB1) in HUVECs clearly demonstrating that *VEAL2* regulates localization and activity of PRKCB2 under in-vivo conditions.

Comment 4: *Were there any survival defects upon manipulation of veal2 or VEAL2 in human ECs? The matrigel tube formation results look very dramatic. What does the monolayer look like? Is it simply a matter of altered adherens junctions and thus permeability or is there also a cell survival phenotype?*

Response) We don't see any major change in monolayer formation in our wound healing assay (presented in Appendix Figure S 24A-D). However, we agree with the reviewer that apoptosis of HUVECs can also lead to a hyper-permeability state. Hence we conducted apoptosis assay in HUVECs under both knockdown and overexpression of *VEAL2* conditions. The lactate dehydrogenase (LDH) based detection of apoptosis in HUVECs under overexpression and knockdown of *VEAL2* conditions showed a lack of any cell toxicity signals. Hence it clearly revealed that

there are no major changes in cell survival in HUVECs and permeability defects largely pertain to altered junctional dynamics. We have provided this data in the revised manuscript. This data is presented in Appendix Figure S **22D** and is given below.

Appendix Figure S 22D: Dot plot representing lactate dehydrogenase levels measured in HUVECs transfected with different overexpression plasmids or siRNAs. Data was acquired from 3 different biological replicates and shown as mean fold change values \pm standard deviation.

Comment 5: *What is the effect of the diabetic conditions in vitro on VEAL2 expression?*

Response) We appreciate the reviewer for this suggestion. We have included this data in our revised manuscript. The stimulation of hyperglycemia in HUVECs by growing cells under high glucose (25mg/ml) resulted in a decrease in the expression of *VEAL2*. It corroborates the findings as observed in choroid tissue of diabetic retinopathy patients. This data is presented in **Expanded figure 5H** and is given below.

Expanded figure 5H: Bar graph representing relative expression of *VEAL2* in control HUVECs cells and hyperglycemia stimulated HUVECs by growing under high glucose. Data was acquired from 3 different biological replicates and shown as mean fold change values \pm standard deviation.

Minor comments:

Comment 1: Does the zebrafish mutant with the 8 basepair deletion (nucleotides 26-33 of the transcript) still make normal levels of the lncRNA?

Response) The allele-specific expression analysis clearly indicated a ~50% reduction in wildtype *veal2* expression in our heterozygous *veal2*^{gib005 Δ 8/+} embryos. We already had data on this, we have included it in our revised manuscript and is presented in **Appendix Figure S 14A**. The data is also given here below.

S14 A

Appendix Figure S 14A: Bar graph representing relative expression of wild type *veal2* allele in control and in *veal2*^{gib005 Δ 8/+} animals showing reduction in expression significantly. Data was acquired from 3 different biological replicates and shown as mean fold change values \pm standard deviation.

Comment 2: The naming convention is problematic since veal2 and VEAL2 are functional homologs, but share no sequence overlap or synteny. Is there a better naming convention that would make it clear that these are not true homologs.

Response) This will be one of the early studies to show functional lncRNA homolog across zebrafish and human without any sequence and structural similarity. We have followed the nomenclature guidelines provided by a recent article published in EMBO (<https://www.embopress.org/doi/full/10.15252/embj.2019103777>). Earlier a study identified and showed homology lncRNA in human and mouse called JPX and Jpx, respectively. They also showed conserved functions of JPX and Jpx despite structural and sequence divergence.

Comment 3: *Line 281 should be 'in vitro' rather than 'in vivo', since this is in cell culture cells.*

Response) We have made this modification in a revised manuscript.

Thank you for submitting a revised version of your manuscript. Your revised study has now been seen by all of the original referees, who now support publication of the revised manuscript. Therefore, I would like to invite you to address the remaining editorial issues before I can extend the official acceptance of the manuscript.

Please let me know if you have any further questions regarding any of these points. You can use the link below to upload the revised files.

Referee #1:

The authors have overall addressed well the comments from the previous round of review, and have improved the manuscript. One comment remains, related to the abundance of the lncRNA in human cells. The authors now report number of copies per microgram RNA. What does this mean per cell? Since micrograms of RNA are typically obtained from a large number of cells, hundreds of copies per microgram does not seem to be "moderate expression" as claimed by the authors. The number should be described and discussed in the context of number of copies per cell, not per microgram.

Referee #2:

The authors addressed many of the comments raised by the reviewers. Although some details may raise specific concerns (the LDH assay is not very sensitive and does not detect apoptosis and the authors could not perform some additional studies due to COVID restrictions), overall the message of the study is sufficiently supported by the data provided. The manuscript therefore can be recommended for acceptance

Referee #3:

The authors have fully responded to the suggestions from the reviewers. The findings are novel, important and highly convincing. Although they have done so because of a reviewer suggestion, I do not think that it is appropriate to plot Ct or deltaCt values for qPCR results (for example, Fig. 3D, Fig. 7A, B, Fig. S21B, etc.). Ct values are not linear values, and it is therefore not appropriate to perform statistics on these values. Ct values are also arbitrary values since they are highly dependent on primer efficiencies. Determining copy numbers using a standard curve is the appropriate way to show absolute quantification. The presentation of the data is also counter-

intuitive, since a higher Ct corresponds to lower expression. In my opinion, fold change is a better way to present this kind of data. Including copy number calculations as supplemental files would be adequate to show how abundant Veal2 transcripts are.

Minor comment: On Line 292, I believe it should be '140 nM' not '140 mM'.

Response to Reviewers Comments:

Referee #1:

The authors have overall addressed well the comments from the previous round of review, and have improved the manuscript. One comment remains, related to the abundance of the lncRNA in human cells. The authors now report number of copies per microgram RNA. What does this mean per cell? Since micrograms of RNA are typically obtained from a large number of cells, hundreds of copies per microgram does not seem to be "moderate expression" as claimed by the authors. The number should be described and discussed in the context of number of copies per cell, not per microgram.

Response: We appreciate the positive feedback from the reviewer. As per reviewer's request we have replotted the absolute quantification data in per cell format. In the original manuscript We had taken 1 T75 flask containing ~0.7M HUVECs. It yielded 30 microgram of total RNA. As per this, the copy number of *VEAL2* per cell would be 0.024 copies. We have presented this data in the revised manuscript and it is reproduced below. Appropriate changes have been made in the revised manuscript (highlighted in yellow).

Figure 5D: Absolute quantification of VEAL2 in HUVECs revealed 0.024 copies cell. 10^2 to 10^8 copies of VEAL2 RNA were used to make the standard curve. Data from three different experiments plotted as mean values \pm standard deviation.

Referee #2:

The authors addressed many of the comments raised by the reviewers. Although some details may raise specific concerns (the LDH assay is not very sensitive and does not detect apoptosis and the authors could not perform some additional studies due to COVID restrictions), overall the message of the study is sufficiently supported by the data provided. The manuscript therefore can be recommended for acceptance

Response: We are pleased to know that the reviewer found our revised manuscript suitable for the EMBO Journal

Referee #3:

The authors have fully responded to the suggestions from the reviewers. The findings are novel, important and highly convincing. Although they have done so because of a reviewer suggestion, I do not think that it is appropriate to plot Ct or deltaCt values for qPCR results (for example, Fig. 3D, Fig. 7A, B, Fig. S21B, etc.). Ct values are not linear values, and it is therefore not appropriate to perform statistics on these values. Ct values are also arbitrary values since they are highly dependent on primer efficiencies. Determining copy numbers using a standard curve is the appropriate way to show absolute quantification. The presentation of the data is also counter-intuitive, since a higher Ct corresponds to lower expression. In my opinion, fold change is a better way to present this kind of data. Including copy number calculations as supplemental files would be adequate to show how abundant Veal2 transcripts are.

Response: We appreciate the positive feedback from the reviewer. We agree with the reviewer's comment that plotting data in fold change would be appropriate and more interpretable. We have re-plotted these data in fold change in the revised manuscript. The following figures have been modified Fig. 3D, 7A, 7B, Extended View fig. EV5G, EV5H, Appendix Fig. S21B to reflect the above mentioned change.

Minor comment:

On Line 292, I believe it should be '140 nM' not '140 mM'.

Response: We have made the correction. Thank you.

Editor accepted the revised manuscript.

Corresponding Author Name: Dr. Sridhar Sivasubbu

Journal Submitted to: The EMBO Journal

Manuscript Number: EMBOJ-2020-107134